



# Particulate organic carbon dynamics in the Gulf of Lion shelf (NW Mediterranean) using a coupled hydrodynamic–biogeochemical model

**Gaël Many**[1], **Caroline Ulses**[1,2], **Claude Estournel**[1,2], **and Patrick Marsaleix**[2]

[1]Laboratoire d'Aérologie, Université de Toulouse, CNRS, UPS, 14 Avenue Edouard Belin, 31400 Toulouse, France
[2]LEGOS, Université de Toulouse, CNES, CNRS, IRD, UPS, 14 Avenue Edouard Belin, 31400 Toulouse, France

**Correspondence:** Gaël Many (gael.many@outlook.fr)

**Abstract.** TS1 CE1 The Gulf of Lion shelf (GoL, NW Mediterranean) is one of the most productive areas in the Mediterranean Sea. A 3D coupled hydrodynamic–biogeochemical model is used to study the mechanisms that drive the particulate organic carbon (POC) dynamics over the shelf. A set of observations, including temporal series from a coastal station, remote sensing of surface chlorophyll $a$, and a glider deployment, is used to validate the distribution of physical and biogeochemical variables from the model. The model reproduces the time and spatial evolution of temperature, chlorophyll $a$, and nitrate concentrations well and shows a clear annual cycle of gross primary production and respiration. We estimate an annual net primary production of $\sim 200 \times 10^4 \, \text{t C yr}^{-1}$ at the scale of the shelf. The primary production is marked by a coast-slope increase with maximal values in the eastern region. Our results show that the primary production is favoured by the inputs of nutrients imported from offshore waters, representing 3 and 15 times the inputs of the Rhône in terms of nitrate and phosphate. In addition, the empirical orthogonal function (EOF) decomposition highlights the role of solar radiation anomalies and continental winds that favour upwellings, and inputs of the Rhône River, in annual changes in the net primary production. Annual POC deposition ($19 \times 10^4 \, \text{t C yr}^{-1}$) represents $10\,\%$ of the net primary production. The delivery of terrestrial POC favours the deposition in front of the Rhône mouth, and the mean cyclonic circulation increases the deposition between 30 and 50 m depth from the Rhône prodelta to the west. Mechanisms responsible for POC export ($24 \times 10^4 \, \text{t C yr}^{-1}$) to the open sea are discussed. The export off the shelf in the western part, from the Cap de Creus to the Lacaze-Duthiers canyon, represents 37 % of the total POC export. Maximum values are obtained during shelf dense water cascading events and marine winds. Considering surface waters only, the POC is mainly exported in the eastern part of the shelf through shelf waters and Rhône inputs, which spread to the Northern Current during favourable continental wind conditions. The GoL shelf appears as an autotrophic ecosystem with a positive net ecosystem production and as a source of POC for the adjacent NW Mediterranean basin. The undergoing and future increase in temperature and stratification induced by climate change could impact the trophic status of the GoL shelf and the carbon export towards the deep basin. It is crucial to develop models to predict and assess these future evolutions.

## 1 Introduction

### 1.1 The importance of continental margins in the organic carbon dynamics

Continental margins are of particular interest concerning the input, production, deposition, and export to the deep open ocean of particulate organic carbon (POC) (Bauer and Druffel, 1998; Liu et al., 2010). These buffer regions often show high biological productivity, induced by solar radiation and nutrient availability from river inputs and coastal upwellings (Legendre, 1990; Dagg et al., 2004; Lohrenz et al., 2008). The input of terrigenous POC and this high productivity make these coastal zones areas of high organic matter de-

position (Gao and Wang, 2008; Dagg et al., 2008). Hydrodynamic processes such as upwelling, dense water cascading, and slope current could favour the lateral transport of POC towards the open sea and deeper environments (Lapouyade and Durrieu de Madron, 2001; Thunell et al., 2007; Sanchez-Vidal et al., 2008). The understanding of the input, deposition, and export of POC is thus essential to estimate the carbon dynamics of coastal areas at a world scale.

In addition, modelling the POC dynamics in coastal and shelf systems needs the integration of several processes interacting with each other, such as a land–sea continuum (riverine organic carbon and nutrient inputs) and hydrodynamical forcings on POC production and transport in the water column. A realistic simulation of the hydrodynamical processes in the coastal area is essential to reproduce the spatiotemporal changes in POC conditions (Hofmann et al., 2011). Among those processes, circulation patterns and stratification dynamics are considered to be extremely important to describe the POC advection, impacted by water mass upwelling, and shelf–open ocean water mass exchanges, as well as to be able to describe the vertical gradients in nutrient conditions that control the POC production (primary production) and the POC deposition (Liu and Chaï, 2009).

## 1.2   Regional settings

The Gulf of Lion shelf (GoL) in the NW Mediterranean is a wide continental shelf (Fig. 1, area of approximately 10 000 km$^2$) strongly influenced by freshwater and particulate matter inputs from the Rhône River (Fig. 1). The Rhône River is characterized by a mean annual discharge of 1700 m$^3$ s$^{-1}$ with annual flood reaching 5000 m$^3$ s$^{-1}$ in autumn and winter, which makes the GoL one of the most river-impacted areas of the Mediterranean (Naudin et al., 1997; Maillet et al., 2006; Ludwig et al., 2009). Sadaoui et al. (2016) estimated a total suspended solid flux around 8.4 × 10$^6$ t yr$^{-1}$ from which approximately 1 % is considered to be POC (see Table 2 in Durrieu de Madron et al., 2000). During floods, terrestrial inputs from the Rhône create a surface plume that spreads southward across the shelf by surface currents driven by continental (westerly/northerly) winds or is constrained along the coast during marine (easterly) winds. In addition, Rhône inputs also feed a bottom nepheloid layer that favours local sediment deposition (Many et al., 2018). The Gulf of Lion is bordered on the continental slope by the Northern Current associated with the cyclonic general circulation of the western Mediterranean basin (Petrenko et al., 2008) intensified in winter (Alberola et al., 1995). The gulf is impacted by strong continental winds, which favour dense water formation and cascading events in winter (Durrieu de Madron et al., 2013) and coastal upwellings in summer (Fraysse et al., 2014). More occasionally, marine storms blowing from the east, particularly in autumn and winter, induce strong along-isobath currents on the shelf, which induce

powerful exports at the southwestern exit of the gulf (Mikolajczak et al., 2020).

In terms of biological net primary production (NPP), the Gulf of Lion is one of the most productive areas in the Mediterranean Sea (together with the North Adriatic and the Alboran seas) (Bosc et al., 2004). It is an exception in this oligotrophic system, which is relatively impoverished in nutrients concerning the open ocean. The annual production in the GoL shelf has been estimated to be in the range of 80–150 g C m$^{-2}$ yr$^{-1}$ (90–165 × 10$^4$ t C yr$^{-1}$ considering a shelf area of 1.1 × 10$^{10}$ m$^2$) (Durrieu de Madron et al., 2000), which is similar to the production in the adjacent deep water formation area (Lefevre et al., 1997; Ulses et al., 2016; Kessouri et al., 2018).

The main mechanisms that drive POC deposition in the Gulf of Lion are widely described in Auger et al. (2011). The authors highlighted the contribution of organic detritus to 80 %–90 % of the total POC deposition, whereas the contribution of living particles (phytoplankton) was estimated to be approx. 10 %–20 %. In addition, the authors estimated that the contribution of terrestrial particulate organic carbon corresponds to less than 17 % of the total of POC deposition, with the main deposition occurring in front of the Rhône mouth during floods. On the other hand, the predominant influence of marine biological production on POC deposition over the entire shelf is highlighted.

The Gulf of Lion is considered a source of POC to the basin of the NW Mediterranean Sea (Durrieu de Madron et al., 2000; Ulses et al., 2008b, 2016). It is a very dynamic system, marked by low residence times (Mikolajczak et al., 2020) where wind-induced currents are important for POC advection. Coastal hydrodynamic conditions are influenced by the circulation along the continental slope, the Northern Current (Petrenko et al., 2003), the freshwater inputs from the Rhône (Marsaleix et al., 1998; Estournel et al., 2001), the wind-driven circulation over the shelf (Estournel et al., 2003; Petrenko, 2003; Petrenko et al., 2005; Ulses et al., 2008a, c), and the formation and cascading of shelf dense water (Dufau-Julliand et al., 2004).

It is however noticeable that the values of the POC dynamics terms have been determined based on local observations and/or during limited periods. The inter-annual variability of the environmental conditions (wind velocity, heat flux, temperature, etc.) as well as episodic events (floods, storms, water mass upwellings) is also expected to play a key role in changes in the POC dynamics over the shelf and needs to be quantified.

Although 3D coupled physical–biogeochemical models are useful tools to analyse the POC dynamics in areas characterized by high spatial and temporal variability, studies based on those models in the Gulf of Lion at pluriannual and shelf scale have remained scarce. Pinazo et al. (1996) investigated the influence of upwelling on primary production on the shelf under various typical wind and Northern Current conditions at a monthly scale, based on a model with a 4 km horizontal

**Biogeosciences, 18, 1–26, 2021**                                           **https://doi.org/10.5194/bg-18-1-2021**

resolution. Tusseau et al. (1998) used a model with a resolution of $1/10°$ ($\sim 11$ km) to estimate primary production and nitrate inputs on the shelf and in particular the shelf–slope exchanges over a year. Auger et al. (2011) analysed and estimated POC deposition at the scale of the shelf over a 4-month period using a 1.5 km resolution model. Campbell et al. (2013) studied the influence of an eddy-induced upwelling on the dynamics of nutrients and phytoplankton using a realistic simulation of the year 2001 with a resolution of 3 km. Based on a coupled simulation of 17 months covering the NW Mediterranean Sea with a horizontal resolution of 1.2 km, Alekseenko et al. (2014) examined the spatial and temporal variability of the stoichiometry of the nutrients and phytoplankton in the NW Mediterranean Sea. Other high resolution (400 m) modelling studies focused on the eastern part of the shelf, in particular the Bay of Marseille, investigating the influence of the Rhône River and Northern Current intrusions on nutrient and phytoplankton dynamics over the period 2007–2011 (Fraysse et al., 2013, 2014; Ross et al., 2016) as well as the variability of the carbonate system in 2007 (Lajaunie-Salla et al., 2021).

### 1.3 Objective of this study

The objective of this present work is to estimate the POC dynamics in the Gulf of Lion shelf and to improve our understanding of the mechanisms that control these dynamics based on a coupled hydrodynamical–biogeochemical model over a pluriannual period and the whole shelf. The 2011–2016 period was characterized by high annual changes in environmental conditions, particularly during winters, which are key periods in the water mass export and mixing and phytoplankton bloom triggering (see winter heat fluxes and winds in Fig. 1). The 2011–2012, 2012–2013, and 2014–2015 periods were marked by cold winters with strong or above-mean heat losses. The 2013–2014 and 2015–2016 periods were characterized by mild winters. The Rhône River discharge was minimal in 2011–2012 and maximal in 2012–2013. At last, winter 2015–2016 was a period characterized by severe marine storms (see the wind rose in Fig. 1). It is expected that these variations in these environmental conditions, which may influence the availability of nutrients in the surface layer, and hence the phytoplankton growth will affect the POC dynamics in the shelf, which remain seldom quantified at the scale of the shelf and during contrasted years.

In this paper, we present the numerical model used to carry out this study, particularly its validation against multiplatform observations including time series from a coastal station, remote sensing of surface chlorophyll *a*, and a glider deployment to describe the vertical distributions of physical and biogeochemical conditions. After describing the environmental conditions, we then estimate the POC dynamics in the shelf during the 2011–2016 period and detail and discuss the variability of the nutrient availability, primary production, the deposition over the shelf and the cross-shelf transport of

POC towards the deep basin and the Catalan margin over this period marked by contrasted meteorological, hydrodynamic, and fluvial conditions.

## 2 Material and methods

### 2.1 The model

The three-dimensional model results from the off-line forcing of the biogeochemical Eco3M-S model by the regional circulation SYMPHONIE model. These two models and the coupling procedure are described thereafter and in the Supplement.

#### 2.1.1 The hydrodynamic model

The SYMPHONIE model (Marsaleix et al., 2006, 2008) is a 3D primitive equation, free surface, and generalized sigma vertical coordinate model. This model was previously used to simulate the hydrodynamic conditions in the Mediterranean Sea and specific processes such as the Rhône River plume dynamics (Estournel et al., 1997; Reffray et al., 2004), coastal dense water formation (Ulses et al., 2008c), wind-induced circulation over the Gulf of Lion shelf (Estournel et al., 2003; Petrenko et al., 2008; Ulses et al., 2008a), shelf–slope exchanges, and along-slope circulation (Bouffard et al., 2008; Mikolajczak et al., 2020).

The numerical grid of the model is the same as in the study of Briton et al. (2018). It consists of a curvilinear bipolar (Bentsen et al., 1999) Arakawa C-grid with 40 vertical sigma levels (Mikolajczak et al., 2020). The bipolar grid presents a horizontal resolution between 300 and 500 m over the shelf and gradually decreases to several kilometres towards the south of the domain along the Algerian coast. This configuration allows us to have more than half of the total points of the grid over the shelf while keeping the open boundaries far from the study area, the Gulf of Lion.

River runoffs were considered using measured daily values for French rivers (Banque Hydro database, http://www.hydro.eaufrance.fr/TS3) and the Ebro (SAIH Ebro database, http://www.saihebro.com/saihebro/index.php TS4) and mean annual values for the others. The implementation of rivers in the model was described in Estournel et al. (2009). Atmospheric forcings were generated by the 3-hourly fields (wind speed and direction, pressure, air temperature and humidity, solar and downward longwave radiation, and precipitation) provided by the ECMWF (European Centre for Medium-Range Weather Forecasts) forecasts. We used the bulk formula of Large and Yeager (2004) to estimate the surface turbulent fluxes.

The period simulated with the SYMPHONIE model runs from 1 July 2011 to 31 December 2016. The initial state and the open boundary conditions were generated by a "parent" simulation (SYMPHONIE) that began 2 months earlier than the "child" simulation. The open lateral boundary con-

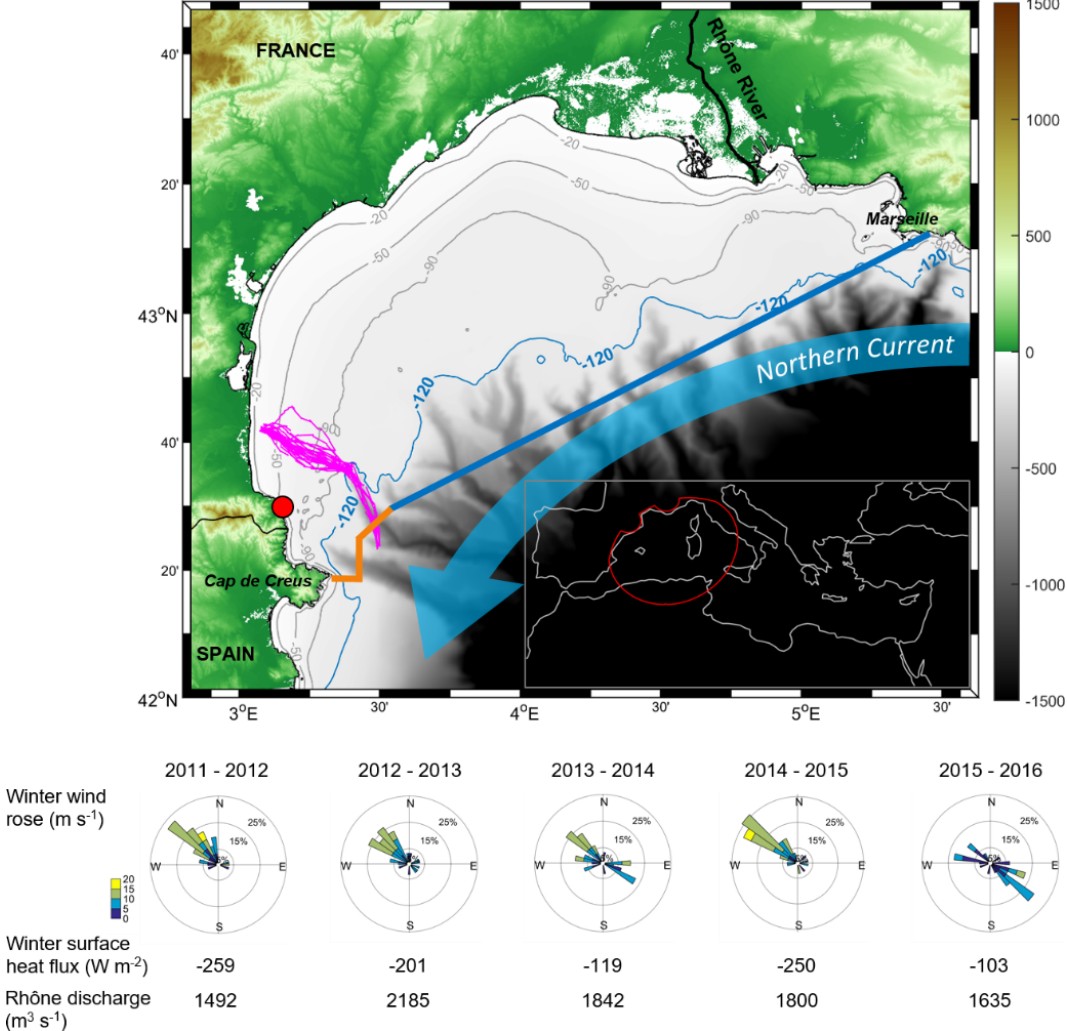

**Figure 1. (a)** Bathymetry (m) of the Gulf of Lion (NW Mediterranean Sea). The Rhône River is shown by a black line. The Northern Current is shown by a thick blue arrow. SOMLIT Banyuls station is shown in red. The path of the glider during the April–May 2013 deployment is shown in magenta. The limit of the shelf studied in this work is specified by the 120 m isobath. The thick blue and orange lines show the limit of the boundaries used to estimate water, nutrients, and particulate organic carbon transports in the eastern and western part of the shelf, respectively. The boundary of the model grid is shown in red on the inserted Mediterranean map. **(b)** Meteorological and fluvial forcings for each year are simulated: the winter (DJF) mean wind rose (m s$^{-1}$), winter surface heat flux (W m$^{-2}$), and annual Rhône River discharge (m$^3$ s$^{-1}$) are specified.

ditions of the child model consist of radiative conditions reinforced by a lateral restoring layer towards the hydrodynamic fields of the parent model (Marsaleix et al., 2006). The parent model covers the Mediterranean basin. Its average horizontal resolution is 4 km. It is initialized with the Mercator Ocean International operational centre hydrodynamic fields using the stratification index correction method described in Estournel et al. (2016).

### 2.1.2 The biogeochemical model

The Eco3M-S model is a multi-plankton and multi-nutrient dynamics model that simulates the dynamics of the bio-geochemical cycles of biogenic elements (carbon, nitrogen, phosphorus, silicon, and oxygen) and plankton groups (Auger et al., 2011; Ulses et al., 2016). This model accounts for seven compartments: a compartment with four dissolved nutrients (nitrate, ammonium, phosphate, silicate), a compartment with three phytoplankton size classes (pico-, nano-, and micro-phytoplankton), a compartment with three zooplankton size classes (nano-, micro-, and meso-zooplankton), a bacteria compartment, a dissolved organic matter compartment, a compartment with particulate organic matter, and a compartment with dissolved oxygen. Figure S1 shows the model structure and flows between the compartments. The model allows a potential multi-nutrient limitation of the phy-

toplankton growth, as it has been observed that phosphorus and nitrogen can both play a controlling role in the Mediterranean Sea (Diaz et al., 2001; Marty et al., 2002). The internal composition, i.e. the stoichiometry, is considered to be flexible for phytoplankton and constant for heterotrophic organisms. A description of the biogeochemical model and its coupling with the hydrodynamical model is given in the Supplement.

Rhône River nutrient inputs (nitrate, ammonium, phosphate, silicate, and dissolved organic carbon) were determined using in situ daily data TS5 (Mistrals-Sedoo database, http://mistrals.sedoo.fr/MOOSE/ TS6 ). Concentrations of dissolved organic phosphorus and nitrogen and particulate organic carbon were estimated from this dataset and the relations found in the literature (Moutin et al., 1998; Sempéré et al., 2000). The Orb, Aude, and Hérault river monthly data were extracted from the Naïades database (Agence de l'eau, http://www.naiades.eaufrance.fr/ TS7 ) and were interpolated on the period of the simulation with a daily resolution. At the other river (Tech, Têt, Agly) mouths, climatological values were used according to Ludwig et al. (2010). The deposition of organic and inorganic matter from the atmosphere was based on the low estimations of Ribera d'Alcala et al. (2003). Finally, the pelagic–benthic coupling of inorganic nutrients was made using the meta-model described in Soetaert et al. (2001). An adjustment of this model was made according to Pastor et al. (2011).

As for the hydrodynamic model, the initial state and the open boundary conditions were generated by a "parent" simulation (Eco3M-S) that encompassed the whole Mediterranean Sea. This latter simulation was forced by the same daily fields from the SYMPHONIE model as used for the "child" regional hydrodynamic model (see Sect. 2.1.1). This ensures the coherence of the physical and biogeochemical fields at the open boundaries of the child regional model. The period simulated with the Eco3M-S regional model runs from 1 August 2011 to 31 July 2016.

### 2.1.3 Estimation of water, nutrients, and POC transport

Water, nutrients, and POC transport are estimated through sections that close off the Gulf of Lion shelf (see Fig. 1). The water column is divided into two parts each time, above and below 60 m corresponding roughly to the depth of the nutricline in summer (Kessouri et al., 2017). The sections are considered down to the bottom with maximum depth depending on the local bathymetry (Fig. 1). The "western" section corresponds to the area known to be responsible for deep export by cascading (sometimes down to the bottom of the basin ∼ 2500 m) during cold winters (Ulses et al., 2008c; Durrieu de Madron et al., 2013). This export is restricted to 300–400 m during mild winters and also during eastern storms, which blow predominantly in autumn and produce a downwelling in the Cap de Creus Canyon (Ulses et al., 2008a;

Mikolajczak et al., 2020). The other section hereafter named "eastern" for the sake of simplicity is known in the eastern part as an intrusion zone of the Northern Current (Conan et al., 1998), while in the centre of the shelf, exchanges with the Northern Current have also been (more rarely) documented (Estournel et al., 2003). It is also the area where the Rhône plume most often exits the shelf under prevailing NW to N wind conditions (Gangloff et al., 2017; Many et al., 2018).

### 2.2 Observations used for the model evaluation

#### 2.2.1 SOMLIT data

Long-term measurements from the Banyuls (42.492° N, 3.153° E) SOMLIT (Service d'Observation en Milieu Littoral) station were downloaded from the SOMLIT website (http://somlit-db.epoc.u-bordeaux1.fr/bdd.php TS8 TS9 ). Daily time series of temperature, salinity, nutrients (nitrate, phosphate), and particulate organic carbon were extracted at the surface (∼ 2 m depth) and close to the bottom (∼ 2 m above the bottom, i.e. ∼ 25 m depth). The description of the data acquisition is detailed in Fraysse et al. (2013) and Liénart et al. (2017, 2018).

#### 2.2.2 Satellite data

Spatial maps of daily chlorophyll-$a$ concentrations, with a 1 km resolution, were obtained using products from the Moderate Resolution Imaging Spectroradiometer (MODIS) (https://oceancolor.gsfc.nasa.gov/ TS10 ). Products, analysis, and calibrations used were provided by IFREMER Nausicaa services and OC5 IFREMER algorithms for chl-$a$ concentrations from Gohin (2011). We then estimated the daily spatial median surface chlorophyll-$a$ concentration (in $\mu g\,L^{-1}$) using a filter to discard images with more than 50 % occupied by clouds over the GoL and discarding surface chlorophyll-$a$ data for depths lower than 20 m since these data could be affected by residual contamination from turbidity despite dedicated treatment.

#### 2.2.3 Glider-based measurements

The glider-based time series (Testor et al., 2018) consist of lines of 25 to 50 km long that run across the shelf from the coast (30 m depth) to the shelf edge (100 m water depth) in the vicinity of the Lacaze-Duthiers canyon head (SW Gulf of Lion) (see glider path in Fig. 1). The autonomous glider was a coastal Teledyne Webb Research Slocum (Davis et al., 2002) that moved at an average speed of 20–30 cm s$^{-1}$ in a sawtooth-shaped trajectory between 1 m below the surface and 1–2 m above the seabed. The glider was equipped with an un-pumped Seabird 41-CP CTD providing temperature, depth, and conductivity data. Salinity was derived following the equation of EOS-80. We then derived the Brunt–Väisälä frequency ($N^2$ expressed in s$^{-2}$), which was used as an indi-

cator of the thermal stratification (see details in Many et al., 2018).

A WetLabs FLNTU sensor provided turbidity (expressed in NTU) and fluorescence of chlorophyll-*a* (factory calibrated and expressed in $\mu g \, L^{-1}$) measurements based on backscattering measurements at 700 nm. Turbidity measurements from the FLNTU ($\lambda = 700$ nm) optical sensor of the glider were used to estimate the particulate backscattering coefficients bbp$_{700}$, which were used to correct fluorescence data from the nonphotochemical quenching (NPQ) (Sackmann et al., 2008; Behrenfeld et al., 2009). The correction applied was determined using the night and day bbp$_{700}$ and fluorescence profiles (see details in Thomalla et al., 2018).

## 3  Model evaluation

### 3.1  Observation–model comparisons at the Banyuls SOMLIT station

Comparisons of simulated surface and bottom temperature and salinity with those measured at the Banyuls SOMLIT station for the period 2011–2016 are presented in Fig. 2. The highly significant correlation (i.e. coefficient of determination $R^2 > 0.8$ for surface data and $R^2 > 0.6$ for bottom data, $p < 0.01$), the RMSD inferior to 0.6, and normalized standard deviation of approx. 1 at the surface suggest that the model reproduces the main changes in physical conditions induced by the variability of heat and water flux and the impact of freshwater discharge.

At the SOMLIT coastal station, the model captures the annual cycle in chl *a*, NO$_3$, PO$_4$, and POC concentrations for surface and bottom waters (Fig. 3). If the model estimates chl-*a* concentrations in summer well, the maximum concentrations in winter–spring are systematically underestimated in the model. The underestimation is more pronounced at the surface than near the bottom. The temporal evolution and magnitude of the modelled nitrate are close to that observed, while the modelled PO$_4$ concentrations were significantly lower in the model than in the observations, in particular near the surface. The discrepancy in PO$_4$ concentration could be explained by the too rapid consumption of this nutrient by phytoplankton in the model. POC concentrations were well estimated in the model (slope of approx. 0.9), which allows the exploitation of the results as part of the POC dynamics estimate.

### 3.2  Surface chlorophyll-*a* comparison between MODIS and the model for the period 2011–2016

The comparison of the daily mean value of the surface chlorophyll *a*, measured from MODIS and extracted from the model at the same points and days, averaged over the GoL shelf is shown in Fig. 4. We obtain mean chlorophyll-*a* concentrations from MODIS and the simulation of 0.39 ($\pm 0.23$) and 0.35 ($\pm 0.24$) $\mu g \, L^{-1}$ over the 2011–2016 period. The re-

lationship between the binned data shows a very good agreement between the model and the observations (slope $= 0.8$; $R^2 = 0.97$; $p < 0.01$) with a mean bias of 0.04 $\mu g \, L^{-1}$. The model reproduces the seasonality of the surface chlorophyll *a* well with the main maximum during the spring period (approx. 1 $\mu g \, L^{-1}$ at the end of March) and a secondary maximum in autumn (approx. 0.6 $\mu g \, L^{-1}$). The spatial patterns with high concentrations in the river plumes were also correctly reproduced (see the bottom panel in Fig. 4). Some discrepancies, however, exist, in particular during spring, where the model could overestimate the surface chlorophyll-*a* concentrations.

### 3.3  Glider–model comparison

The comparison between the data from the glider deployment in April–May 2013 and those extracted from the model at the same time and position is shown in Fig. 5. Overall, the comparison shows a good agreement between the descriptions of temperature, salinity, and chlorophyll-*a* conditions in the model and glider data, with mean biases of 0.33 °C, 0.17, and 0.001 $\mu g \, L^{-1}$, respectively. This period was characterized by the establishment of the water column stratification. Temperature, salinity, and Brunt–Väisälä frequency derived from the glider data reflect the vertical stratification that controlled the chlorophyll-*a* vertical distribution. The model accurately reproduces these vertical thermal and chlorophyll-*a* distributions, although some differences exist, such as the intensity of the stratification (see the Brunt–Väisälä frequency in Fig. 5). Chlorophyll-*a* measurements along the glider path in April–May 2013 indicate frequent occurrences of subsurface chlorophyll-*a* maxima (approx. 1.5 $\mu g \, L^{-1}$) that is well represented in the model data despite an underestimation of the intensity (see bottom panel in Fig. 5).

## 4  Results

### 4.1  Annual cycles and estimates of physical and biogeochemical conditions

#### 4.1.1  Atmospheric, hydrodynamic, and fluvial conditions

Time series of the simulated surface solar radiation, heat flux, and stratification index are shown in Fig. 6 (spatially averaged over the GoL shelf). The stratification index is estimated as the vertical integration of density profiles along depth (expressed in kg m$^{-2}$) and then spatially averaged over the shelf. It represents the amount of buoyancy to be extracted to mix the water column from the surface to the bottom and achieve a homogenous density equal to the bottom density. Figure 6b shows that the shelf of the Gulf of Lion lost heat at the air–sea interface from October to mid-March and gained heat from the atmosphere from mid-March to September. Heat flux shows a strong interannual varia-

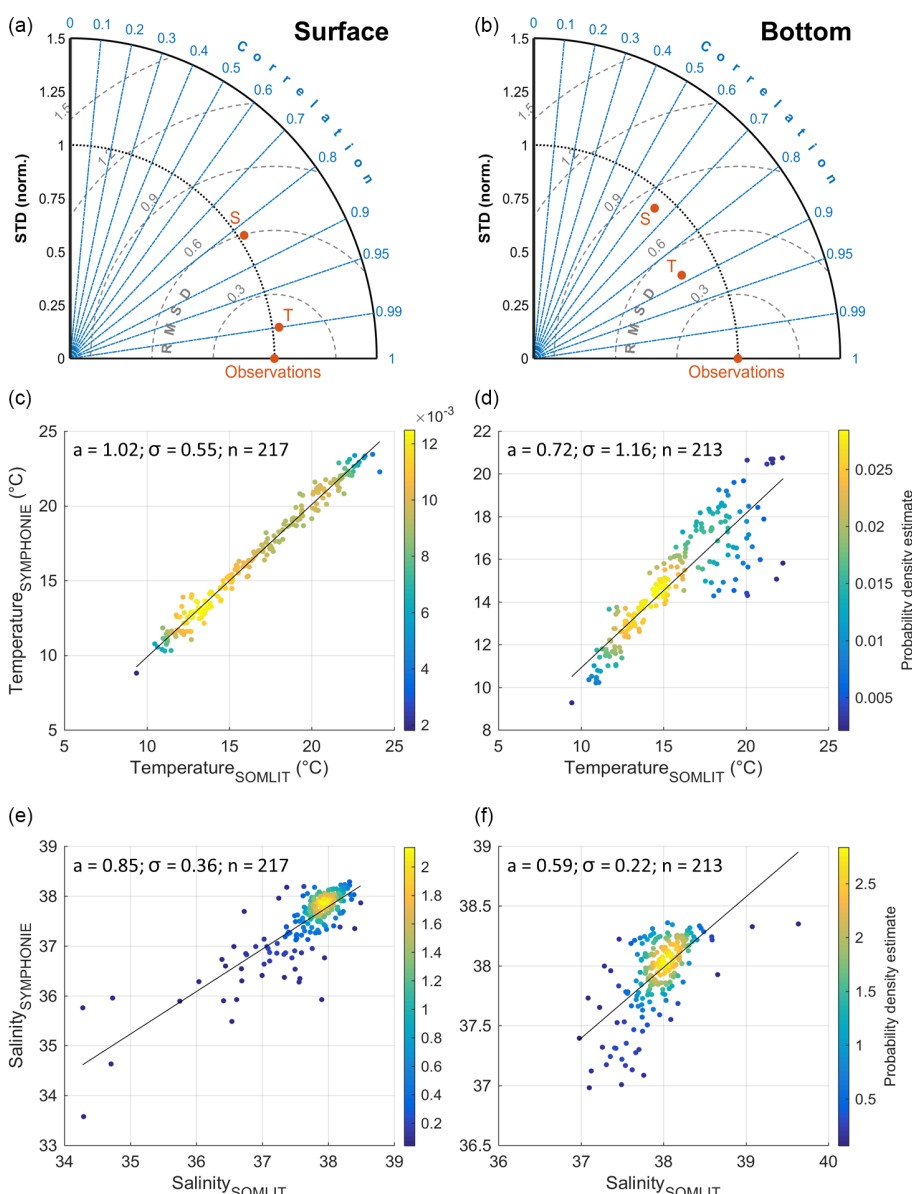

**Figure 2.** Comparison of simulated and measured (SOMLIT) temperature ($T$ in top panels) (°C) and salinity ($S$ in **a, b**) TS11 at the surface (**a**) TS12 and bottom (**f**) TS13 layers for the period 2011–2016. The scatter plots show the density of points (i.e. the kernel density estimation of simulation–observation pairs). The slope of the relation (**a**), the standard deviation ($\sigma$), and the number of data ($n$) is specified.

tion between the end of November and mid-February when strong heat loss events occurred, reaching a monthly average of 400 W m$^{-2}$ in February 2012. The stratification of the water column also exhibited a clear annual cycle (Fig. 6c). From October to early December the cold northerly wind events induced a progressive decrease in the stratification. It increased from March until July when the shelf warmed up. The interannual variability appeared quite strong in summer and autumn.

Volume transport is assessed in Fig. 7 through two unequal sections that close off the Gulf of Lion shelf (see Fig. 1, details in Sect. 2.1.3). From the end of spring to summer (May–September), the exchanges between the coast and the open sea resulted in an import to the west and export to the east limited to the surface layer and of the order of 0.1 Sv on average (Fig. 7a). In autumn (usually October and/or November, brown shaded area in Fig. 7), the shelf imported waters from the east and exported waters to the west, with exchanges on both layers between 0.1–0.2 Sv. In late winter and early spring (roughly from February to April, but especially in 2012 and 2013) the exchanges were in the same direction but took place mostly in the deep layer, reaching 0.2 Sv in February 2012 (Fig. 7b).

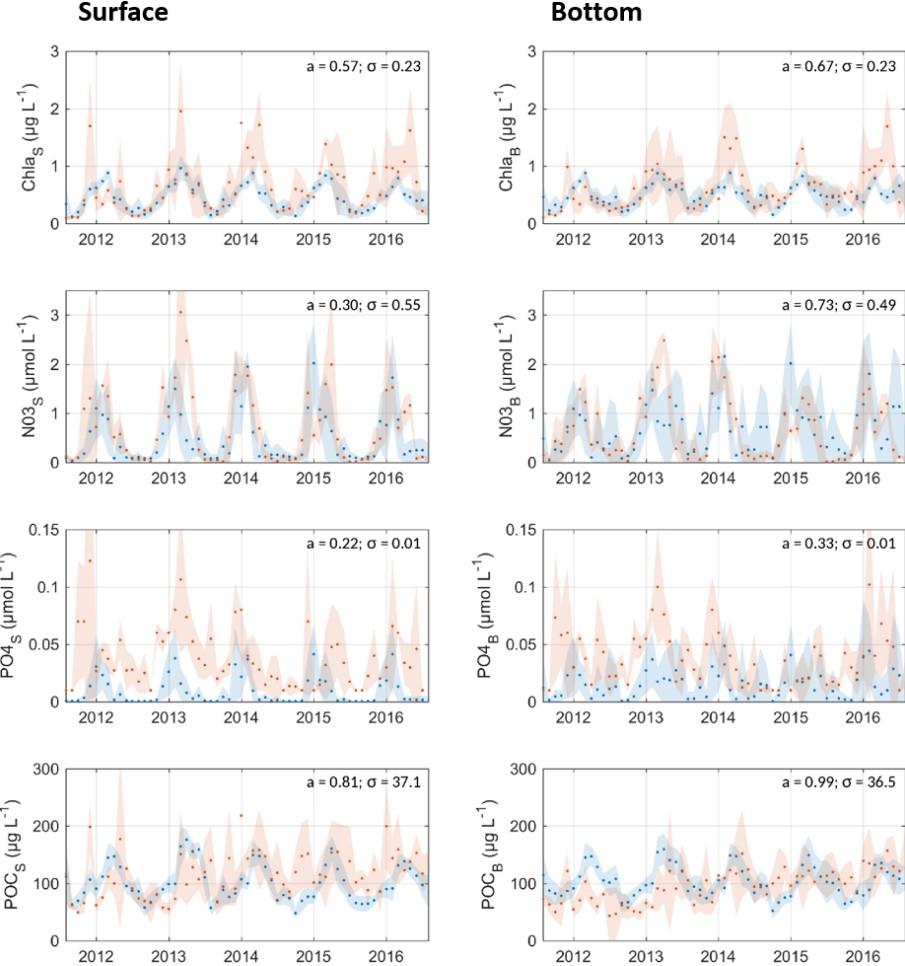

**Figure 3.** Comparison of monthly averaged simulated (blue) and measured (orange – SOMLIT Banyuls station) concentration of (from top to bottom) chl $a$ (mug chl $a$ L$^{-1}$), NO$_3$ (µmol N L$^{-1}$), PO$_4$ (µmol P L$^{-1}$), and POC (µg C L$^{-1}$) from the surface (left) and bottom (right) waters. Standard deviations are shown by shaded areas. The slope of the linear relation model to observation (a) TS14 and the mean standard deviation ($\sigma$) are specified. Note that here the bacteria and mesozooplankton concentrations are excluded from the POC calculation to fit with the measurement method.

### 4.1.2 Nutrients and phytoplankton

*External inputs of inorganic nutrients.* Cross-shelf transport of NO$_3$ (similar results are observed for PO$_4$ and are not shown) computed along the GoL shelf boundary (the boundary is indicated in Fig. 1) shows the import of nutrients year-round from adjacent seas ($22.8 \pm 2.3 \times 10^4$ t N yr$^{-1}$ and $2.92 \pm 0.30 \times 10^4$ t P yr$^{-1}$) (see yellow lines in Fig. 8 and details in Table 1). Nutrients were mainly imported from offshore through the deep layer on the eastern part of the shelf and year-round (Fig. 8b) with a maximum in autumn and winter ($4$–$6 \times 10^4$ t N per month), especially in 2012, 2013, and 2015. These imports in the deep layer of the shelf were much stronger than the exports, which took place mainly in autumn and winter and to the west while they were almost zero in spring and summer. In the surface layer, imports and exports were low year-round except in winter for export,

which then was of the same order or even higher than in the deep layer. In addition, strong inputs from rivers took place in autumn, winter, and spring (not shown). The Rhône inputs represented 95 % to 96 % of the total river inputs. Nitrate river inputs are estimated to be $7.1 \pm 1.3 \times 10^4$ t N yr$^{-1}$ with maximum values in 2012–2013 (Table 1). Phosphate river inputs are estimated to be $0.19 \pm 0.04 \times 10^4$ t P yr$^{-1}$ (max. in 2012–2013, Table 1).

*Inventories over the shelf.* Figure 9a–c show the annual cycle of nutrient and phytoplankton inventories integrated over the water column on the Gulf of Lion shelf (bathymetry < 120 m) and for the 5 years studied. Figure 9d–f show the annual cycle of the nutrient and chlorophyll-$a$ profiles averaged on the Gulf of Lion shelf and over the period 2011–2016. The NO$_3$ inventories over the shelf showed a mean annual value of $2.2 \times 10^4$ t N and 14 % of inter-annual variability. Regarding the PO$_4$, the shelf showed a mean an-

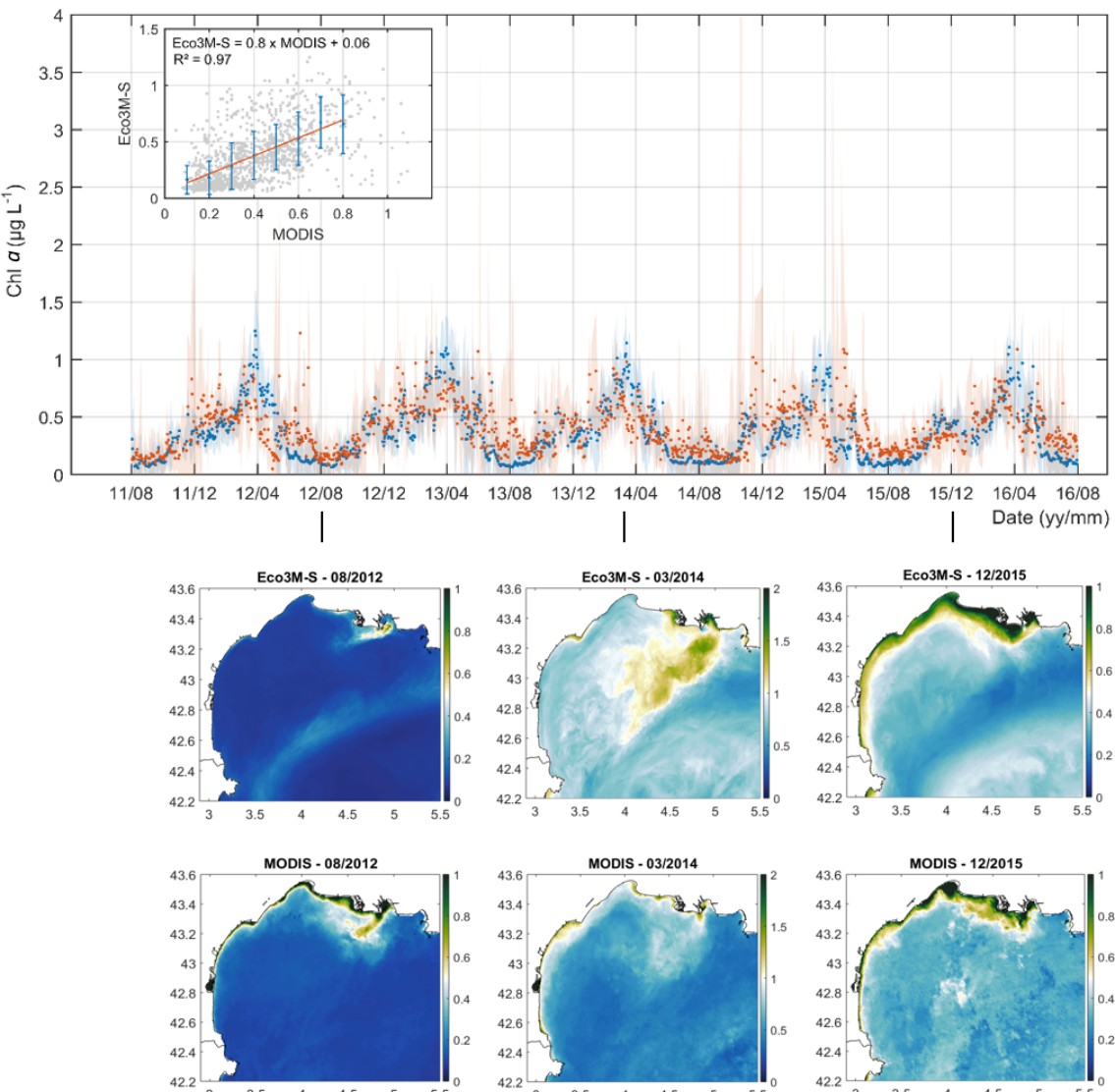

**Figure 4.** The top shows a comparison of the daily mean value of surface chlorophyll $a$ ($\mu g\,L^{-1}$) from MODIS (orange) and extracted from the Eco3M-S model (blue). The linear relationship is shown in the top-left corner. Standard deviations of each estimate are shown by shaded areas. The bottom shows comparisons of monthly mean surface chlorophyll $a$ (in $\mu g\,L^{-1}$) from Eco3M-S (top) and MODIS (bottom) in summer 2012, spring 2014, and winter 2015.

**Table 1.** Annual average nutrient inventories ($10^4$ t N and t P) and annual external inputs ($10^4$ t N and t P yr$^{-1}$) from the rivers (percent of the Rhône is detailed) and from offshore (by convention positive values show an import of nutrients).

|  | 2011–2012 | 2012–2013 | 2013–2014 | 2014–2015 | 2015–2016 | MEAN | SD |
|---|---|---|---|---|---|---|---|
| $NO_3$ inventory | 1.9 | 2.6 | 2.3 | 2.1 | 2.0 | 2.2 | 0.3 |
| $NO_3$ river input (%Rhône) | 5.8 (97) | 9.2 (96) | 7.2 (96) | 6.6 (94) | 6.5 (98) | 7.1 (96) | 1.3 |
| $NO_3$ transport from adjacent seas | 26.6 | 22.8 | 22.6 | 20.6 | 21.6 | 22.8 | 2.3 |
| $PO_4$ inventory | 0.15 | 0.18 | 0.16 | 0.14 | 0.13 | 0.15 | 0.02 |
| $PO_4$ river input (%Rhône) | 0.15 (95) | 0.25 (95) | 0.20 (96) | 0.18 (93) | 0.15 (96) | 0.19 (95) | 0.04 |
| $PO_4$ transport from adjacent seas | 3.39 | 2.98 | 2.91 | 2.66 | 2.64 | 2.92 | 0.30 |

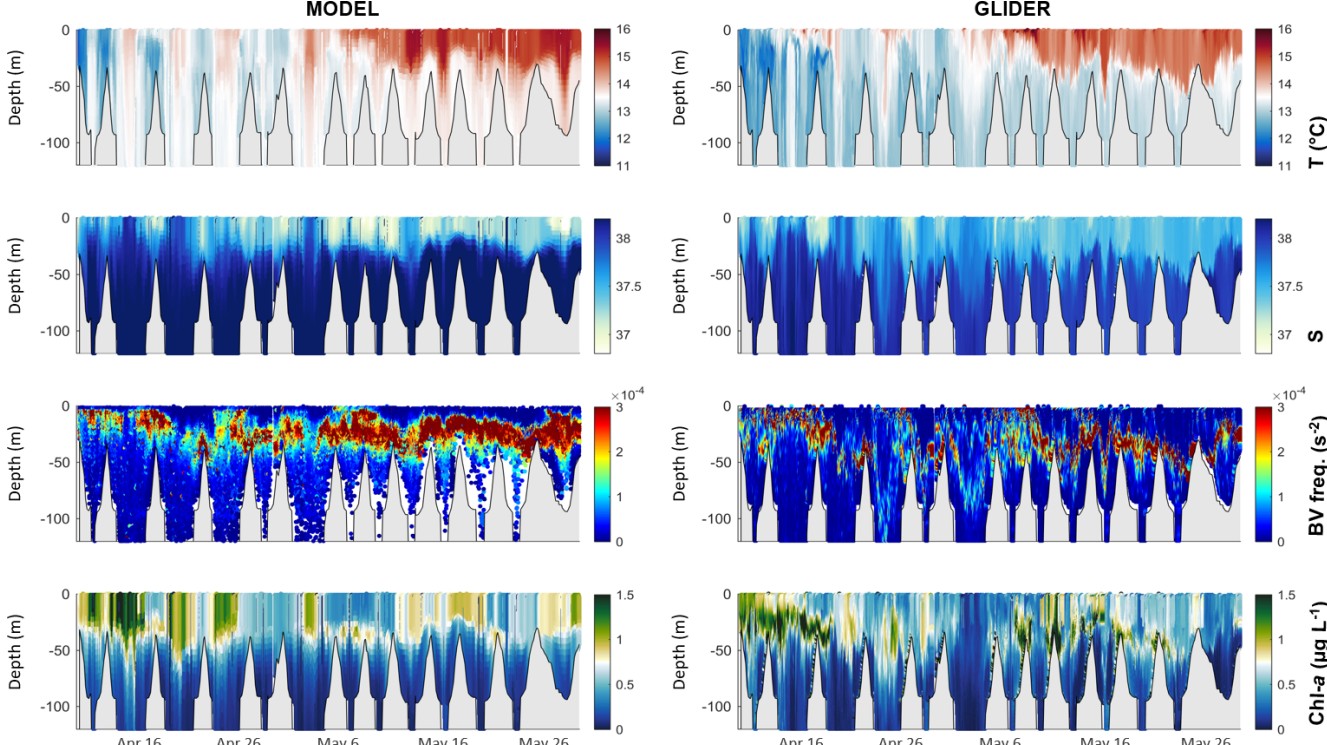

**Figure 5.** Comparisons of model outputs (left) and glider-based measurements (right) of (from top to bottom) temperature (°C), salinity, derived Brunt–Väisälä frequency (s$^{-2}$), and chlorophyll $a$ (µg L$^{-1}$, corrected from quenching for glider data). Simulated data correspond to the extracted data at the glider time and position. Bathymetry is shown in grey.

nual value of $0.15 \times 10^4$ t P and 13 % of inter-annual variability (Table 1). The nutrient inventories were minimum in summer during the stratified period (Fig. 9a–b). The upper layer was depleted in nutrients, and the nutriclines were located at $\sim 60$ m depth (Fig. 9d–e). A deep chlorophyll-$a$ maximum (DCM) with concentrations of $0.5$ mg m$^3$ was present between 40 and 60 m depth (Fig. 9f). From September onwards, events of vertical mixing associated with northerly gales led to injections of nutrients into the upper layer through the nutricline (Fig. 9a–b) and erosion of the DCM.

In November–December, nutrient inventories increased sharply (Fig. 9a–b), and nutrient profiles became homogeneous over the water column (Fig. 9d–e). Nutrient inventories reached their maximum between the end of December and February when a significant inter-annual variation is found. During this winter period with minimal solar radiation (Fig. 6a), the vertical mixing drove the phytoplankton cells downward where the light intensity was low (Fig. 9f). The phytoplankton biomass was then minimal (Fig. 9c). Superimposed on this seasonal feature, at the monthly scale, significant decreases in chlorophyll-$a$ inventories were observed in some years in November and December, clearly linked to periods of low solar radiation. From February onwards, as solar radiation increased again, the model predicted a strong increase in phytoplankton biomass in the upper layer and a de-

crease in nutrient inventories. The chlorophyll-$a$ concentration reached a maximum in March–April (value between 0.8 and 1.3 mg m$^{-3}$) when the water column restratified (Fig. 9f). In April, DCM reformed when nutrients began to be depleted in the surface layer. It gradually deepened with the deepening of the nutriclines. The phytoplankton biomass decreased.

### 4.1.3 POC fluxes

The time series of the simulated monthly POC fluxes for the 5 years is shown in Fig. 10 (spatially integrated over the GoL shelf). Related annual estimates of physical (cross-shelf transport and deposition) and biogeochemical POC fluxes are synthesized in Table 2. GPP (gross primary production) was maximum during spring and summer ($\sim 15\,000$ t C d$^{-1}$) and decreased in winter ($\sim 5000$ t C d$^{-1}$) (Fig. 10a). A highly significant correlation of 0.87 ($p < 0.01$) is found between GPP and solar radiation. The increase in solar radiation yielded the onset of the late winter–spring GPP (Fig. 10a). Overall, we estimate the POC produced through the GPP at the scale of the shelf to be $408.7 \pm 4.9 \times 10^4$ t C yr$^{-1}$.

The total community respiration, which corresponds to the transformation of POC to dissolved inorganic carbon (DIC) by phytoplankton, zooplankton, and bacteria, followed the pattern of the GPP with maximum values during late spring and summer (Fig. 10b). In detail, au-

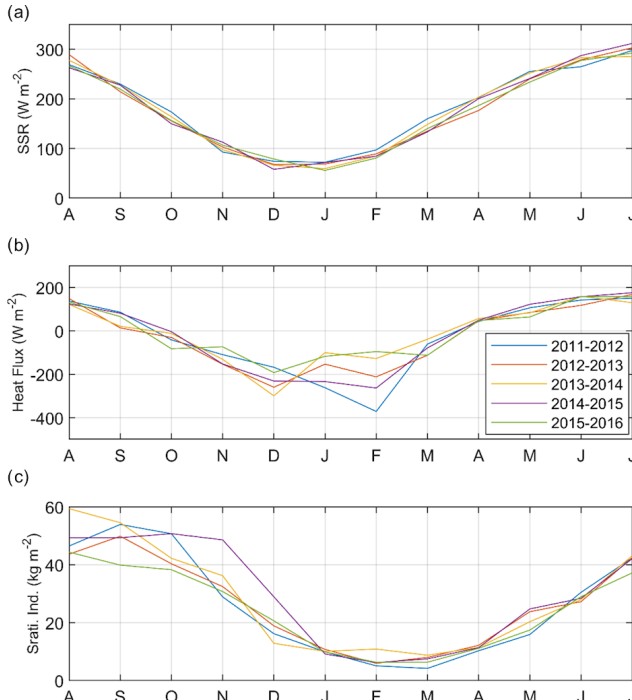

**Figure 6.** Time series for the 5 simulated years of monthly **(a)** surface solar radiation (W m$^{-2}$), **(b)** heat flux (W m$^{-2}$), and **(c)** stratification index (kg m$^{-2}$).

totrophic respiration ($88.6 \pm 3.1 \times 10^4$ t C yr$^{-1}$), in addition to the exudation ($121.2 \pm 8.2 \times 10^4$ t C yr$^{-1}$), led to a quantity of POC degraded or recycled in the water column by the producers of about $209.8 \pm 6.6 \times 10^4$ t C yr$^{-1}$ (51 % of the GPP). This entailed a net primary production (NPP) of $198.9 \pm 8.9 \times 10^4$ t C yr$^{-1}$. The NPP followed the same patterns during all years with minimum productivity in winter ($\sim 3000$ t C d$^{-1}$) and maximum during bloom onset in spring ($\sim 8000$ t C d$^{-1}$ maximum in April 2012 and June 2013) (Fig. 10e). In addition, $229.5 \pm 5.4 \times 10^4$ t C yr$^{-1}$ of OC was remineralized by the heterotrophic respiration (79 % from bacteria activity). The net ecosystem production (NEP, Fig. 10c), which is the difference between the GPP and the total community respiration, shows that the ecosystem was productive overall, with NEP estimates of about $90.6 \pm 3.3 \times 10^4$ t C yr$^{-1}$ at the scale of the shelf (Table 2). The other annual mean DOC and POC flows in the model were estimated at $69.2 \pm 4.1 \times 10^4$, $127.1 \pm 3.4 \times 10^4$, $6.7 \pm 0.1 \times 10^4$, and $259.6 \pm 6.3 \times 10^4$ t C yr$^{-1}$ for messy feeding, decomposition of POC to DOC, bacteria mortality, and DOC uptake, respectively (Table 2). POC fluxes from river inputs were highly variable in time and mainly related to floods of the Rhône River, which locally brought more than $5000$ t C d$^{-1}$ during episodic events (Fig. 10d). Overall, it yielded a quantity of POC delivered from rivers of about $13.6 \pm 9.8 \times 10^4$ t C yr$^{-1}$ with a large inter-annual variability of 72 %. It is noticeable

that while the concentration of POC in the river decreases during floods, the important volume of water delivered during such events considerably increases the input of POC to the shelf.

Cross-shelf transport of POC computed along the GoL shelf boundary (Fig. 10f) was highly variable in time and oriented off the GoL shelf with maximum values in winter and spring (max. export of $2000$ t C d$^{-1}$ in May 2012 and March and June 2013). During summer and autumn, the net transport was weaker. Annually, this led to a net total value of about $24.0 \pm 4.2 \times 10^4$ t C yr$^{-1}$ of POC exported towards the open sea and the Catalan margin (see details in Sect. 4.4).

At last, POC fluxes from the rivers and the biological activity highly contributed to POC deposition over the shelf (Fig. 10g, correlation of $R = 0.53$ and $R = 0.52$ ($p < 0.001$), respectively). The 5 years show the role of the NPP on the temporal background with POC deposition between 400 and $800$ t C d$^{-1}$ over the shelf. In addition, episodic inputs from rivers during floods increased POC deposition to more than $800$ t C d$^{-1}$ at the scale of the shelf in May and June 2013. The different contributions to the POC deposit led to a total of about $26.9 \pm 6.3 \times 10^4$ t C yr$^{-1}$.

### 4.2 Spatial variability of the annual NPP over the shelf and interannual variability

The vertically integrated NPP for the 2011–2016 period simulated is presented in Fig. 11. The NPP was not uniform over the shelf. Minima were located along the coast for depths lower than 50 m and showed values in the range of 50–100 g C m$^{-2}$ yr$^{-1}$. Further offshore, the NPP increased to 150–200 g C m$^{-2}$ yr$^{-1}$ all over the shelf with maximum values close to the shelf break (120 m depth, $\sim 200$ g C m$^{-2}$ yr$^{-1}$). While a trend exists between the NPP and the depth, local maxima were located at the eastern entrance of the shelf and in front of the Rhône mouth with values of approx. $250$ g C m$^{-2}$ yr$^{-1}$ in the ROFI (region of freshwater influence).

The annual anomalies (Fig. 11) show that the first 2 years (2011–2012 and 2012–2013) of the simulation were more productive than the rest of the simulated years with mean NPP anomalies of $+9$ g C m$^{-2}$ yr$^{-1}$. Conversely, the 2013–2016 period showed a lower NPP with mean anomalies of $-7$ g C m$^{-2}$ yr$^{-1}$. In detail, our results show that there are maximum anomalies in particular areas of the Gulf of Lion, namely inside the Rhône ROFI, along the coast, and over the southwestern external part of the shelf.

### 4.3 Spatial variability of the annual POC deposition over the shelf

The POC deposition over the shelf averaged over the 5 years simulated and its annual anomalies are presented in Fig. 12. Our estimates yielded an averaged POC deposition of 19.3 g C m$^{-2}$ yr$^{-1}$. As for the NPP, the deposition of POC

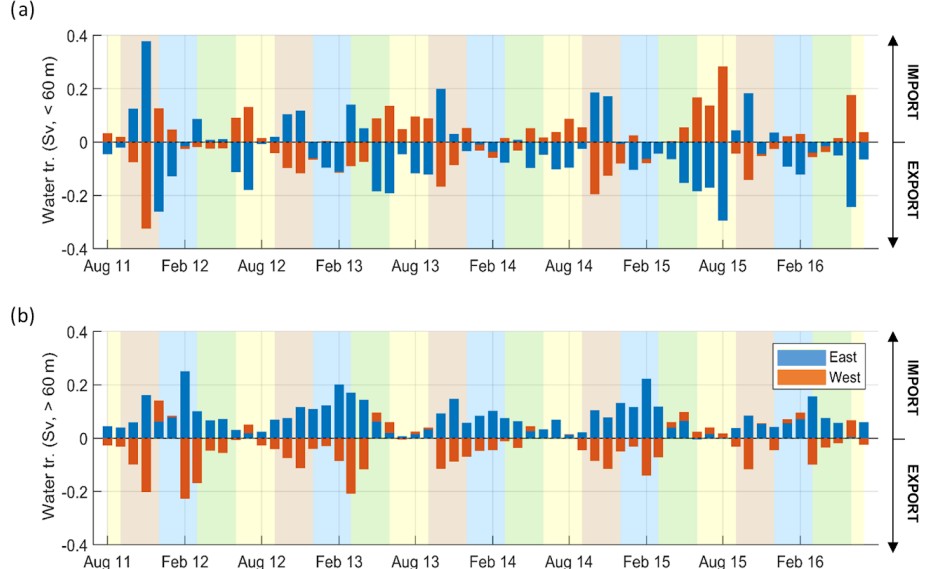

**Figure 7. (a)** Time series for the 5 simulated years of monthly water transport in the eastern and western parts of the shelf (in Sv – by convention export off the shelf is shown by negative values, and the shelf boundaries are indicated in Fig. 1) at the surface ($< 60\,\text{m}$ depth). Shaded areas show the different seasons over the period simulated (yellow, JJA; brown, SON; blue, DJF; green, MAM). **(b)** Same as **(a)** but for depths superior to $60\,\text{m}$.

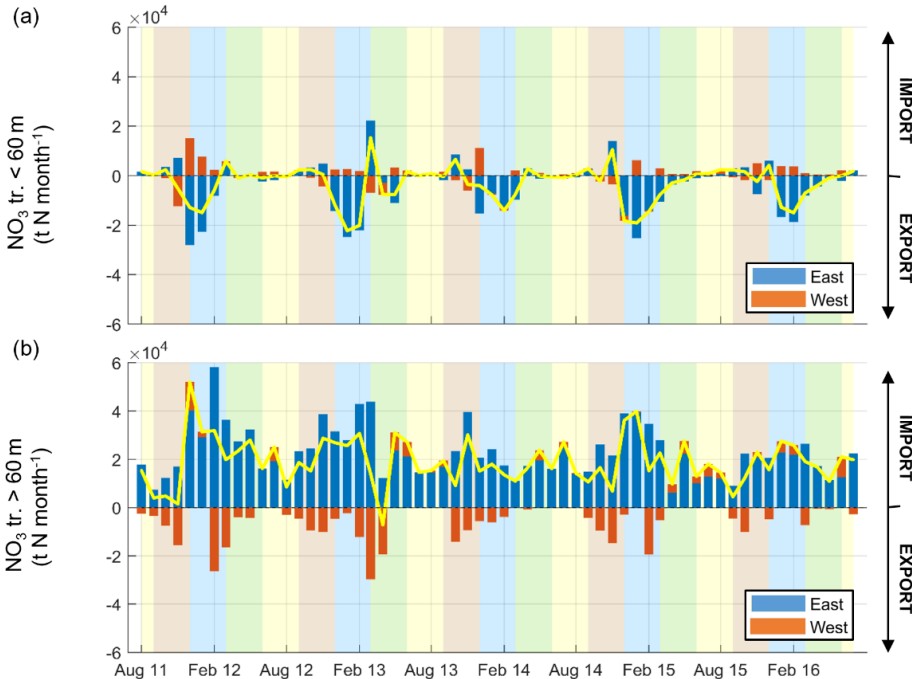

**Figure 8. (a)** Temporal variability of the monthly net surface (depths $< 60\,\text{m}$) transport of $NO_3$ (t N per month) through the sections defined in Fig. 1. By convention, import (export) of $NO_3$ is shown by positive (negative) values. The residual net transport is shown by the yellow line. Shaded areas show the different seasons over the period simulated (yellow, JJA; brown, SON; blue, DJF; green, MAM). **(b)** Same as **(a)** but for depths $> 60\,\text{m}$.

Biogeosciences, 18, 1–26, 2021                                                                    https://doi.org/10.5194/bg-18-1-2021

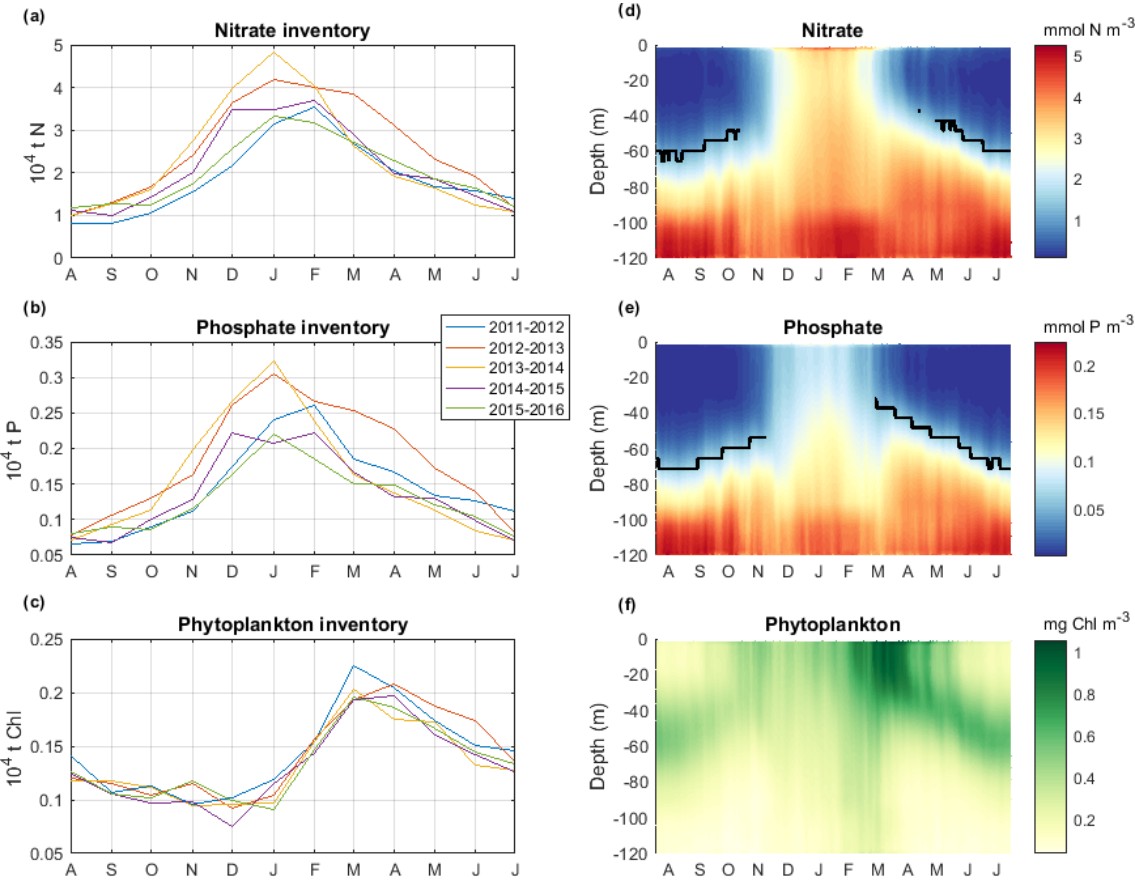

**Figure 9.** Time series for the 5 simulated years of depth-integrated monthly averaged inventory of **(a)** nitrate ($10^4$ t N), **(b)** phosphate ($10^4$ t P), and **(c)** chlorophyll $a$ ($10^4$ t chl $a$) and for the climatological vertical section of daily concentrations of **(d)** nitrate (mmol N m$^{-3}$), **(e)** phosphate (mmol P m$^{-3}$), and **(f)** chlorophyll $a$ (mg chl $a$ m$^{-3}$). Black lines in **(e)** and **(f)** represent nutriclines (1 mmol N m$^{-3}$ and 0.05 mmol P m$^{-3}$).

was not uniform over the shelf. It was maximum in front of the river mouths, in particular the Rhône river mouth, with values of about 30 g C m$^{-2}$ yr$^{-1}$. The POC deposit decreased from 20 g C m$^{-2}$ yr$^{-1}$ in the middle of the shelf, between 20 and 60 m depth, to 10 g C m$^{-2}$ yr$^{-1}$ at the shelf break.

The annual anomalies (Fig. 12) show that during the first two years (2011–2012 and 2012–2013) of the simulation POC deposition was higher than the rest of the simulation with mean anomalies of +0.9 and +0.3 g C m$^{-2}$ yr$^{-1}$. Conversely, the 2013–2016 period showed a lower POC deposition with a mean anomaly of −0.5 g C m$^{-2}$ yr$^{-1}$. In detail, the model shows that there were maximum anomalies in particular areas of the Gulf of Lion shelf, namely over the Rhône prodelta, along the coast for depths lower than 50 m, and to a lesser extent in the central area of the shelf.

### 4.4 Cross-shelf transport of POC

Over the 2011–2016 period, the annual net POC transport off the shelf was estimated to be $24.0 \times 10^4$ t C yr$^{-1}$. We detailed the transport through different sections of the GoL shelf boundary shown in Fig. 13.

Results highlight the preferential area of the southwestern part of the Gulf of Lion shelf (approx. 10 % of the total boundary), which corresponded to 37 % of the total POC net transport off the shelf considering the whole water column ($8.9 \times 10^4$ t C yr$^{-1}$). Within this western section, the transport was carried out mainly through the Cap de Creus Canyon (CC in Fig. 13, 18 % of the total export) and towards the Catalan shelf to the south of the GoL (CS in Fig. 13, 19 % of the total export). In addition, our results also show the balanced net transport in the Lacaze-Duthiers canyon (LD in Fig. 13).

Considering only surface waters (0–60 m depth), the POC net transport, oriented towards the open sea and adjacent shelf, amounted to $13.6 \times 10^4$ t C yr$^{-1}$, 57 % of the total export, and mainly occurred in the eastern and central parts of the shelf.

The temporal variability of the POC net transport is shown in Fig. 14. The results show the variability of the export of POC to the open sea through the surface layer (Fig. 14a,

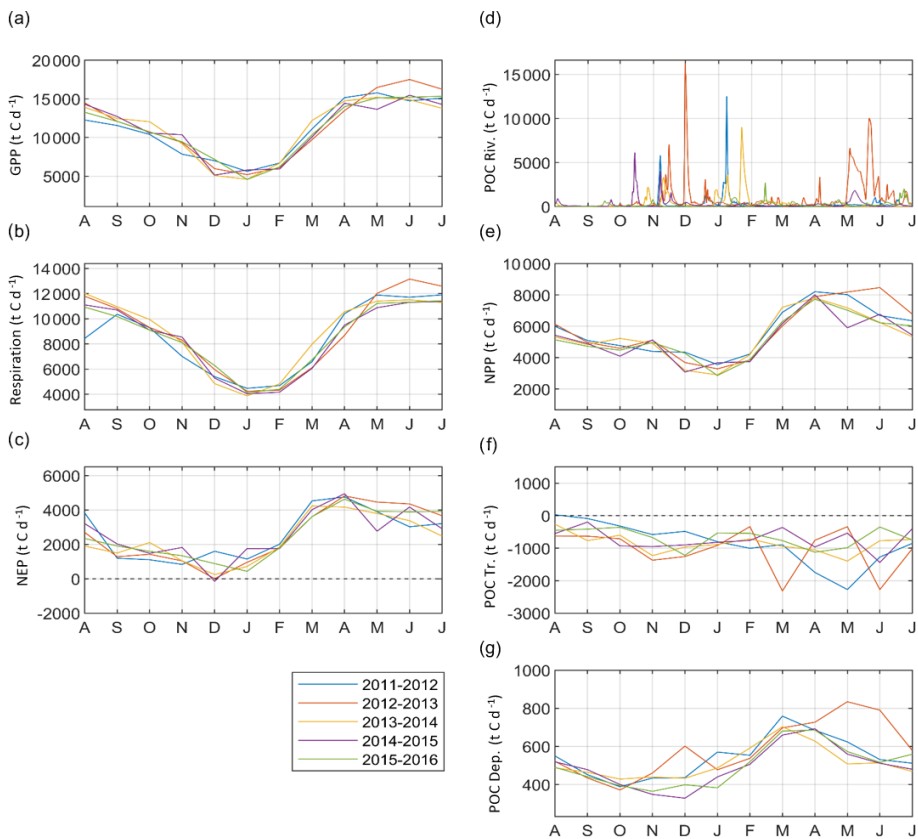

**Figure 10.** Time series for the 5 simulated years of POC fluxes (in $t\,C\,d^{-1}$) from **(a)** monthly gross primary production, **(b)** monthly total respiration, **(c)** monthly net ecosystem production (NEP), **(d)** daily river input, **(e)** monthly net primary production, **(f)** monthly transport across the shelf boundary, and **(g)** monthly deposition.

**Table 2.** Annual average of POC inventory (in $10^4\,t\,C$) and annual POC fluxes (in $10^4\,t\,C\,yr^{-1}$) in the Gulf of Lion shelf. Cross-shelf net transport across the Gulf of Lion's shelf boundary (indicated in Fig. 1) is specified. By convention net transport off the Gulf of Lion's shelf is shown by a negative value. Total community respiration: pelagic autotrophic and heterotrophic respiration; Reminealisation$_{Sed}$: benthic remineralization.

| | 2011–2012 | 2012–2013 | 2013–2014 | 2014–2015 | 2015–2016 | MEAN | SD |
|---|---|---|---|---|---|---|---|
| POC inventory | 7.9 | 8.0 | 7.7 | 7.3 | 7.5 | 7.7 | 0.3 |
| GPP | 409.9 | 415.3 | 410.0 | 402.0 | 406.5 | 408.7 | 4.9 |
| Exudation | 109.4 | 115.9 | 126.5 | 128.5 | 125.9 | 121.2 | 8.2 |
| NPP | 209.3 | 207.0 | 195.7 | 188.8 | 193.6 | 198.9 | 8.9 |
| Total community       respiration | 315.9 | 326.7 | 324.1 | 310.1 | 313.7 | 318.1 | 7.0 |
| Heterotrophic respiration | 224.7 | 234.4 | 236.4 | 225.4 | 226.7 | 229.5 | 5.4 |
| Autotrophic respiration | 91.2 | 92.3 | 87.7 | 84.7 | 87.0 | 88.6 | 3.1 |
| NEP | 94.0 | 88.6 | 85.9 | 91.9 | 92.8 | 90.6 | 3.3 |
| Messy feeding | 70.3 | 75.0 | 70.0 | 64.7 | 66.0 | 69.2 | 4.1 |
| Decomposition of POC to DOC | 128.3 | 131.8 | 127.9 | 124.7 | 122.9 | 127.1 | 3.4 |
| Bacteria mortality | 6.7 | 6.5 | 6.7 | 6.8 | 6.6 | 6.7 | 0.1 |
| Uptake of DOC by bacteria | 251.6 | 259.9 | 269.3 | 258.3 | 259.1 | 259.6 | 6.3 |
| Rivers             (% Rhône) | 9.1 (96) | 30.8 (98) | 12.4 (97) | 8.6 (94) | 7.0 (97) | 13.6 (97) | 9.8 |
| Deposition | 25.0 | 38.0 | 25.9 | 23.3 | 22.5 | 26.9 | 6.3 |
| Reminealisation$_{Sed}$ | 11.7 | 14.4 | 14.9 | 14.2 | 14.0 | 13.8 | 1.3 |
| Buried$_{Sed}$ | 13.3 | 23.6 | 11.0 | 9.2 | 8.5 | 13.1 | 6.1 |
| Cross-shelf net transport | −25.7 | −30.0 | −24.4 | −21.0 | −19.2 | −24.0 | 4.2 |

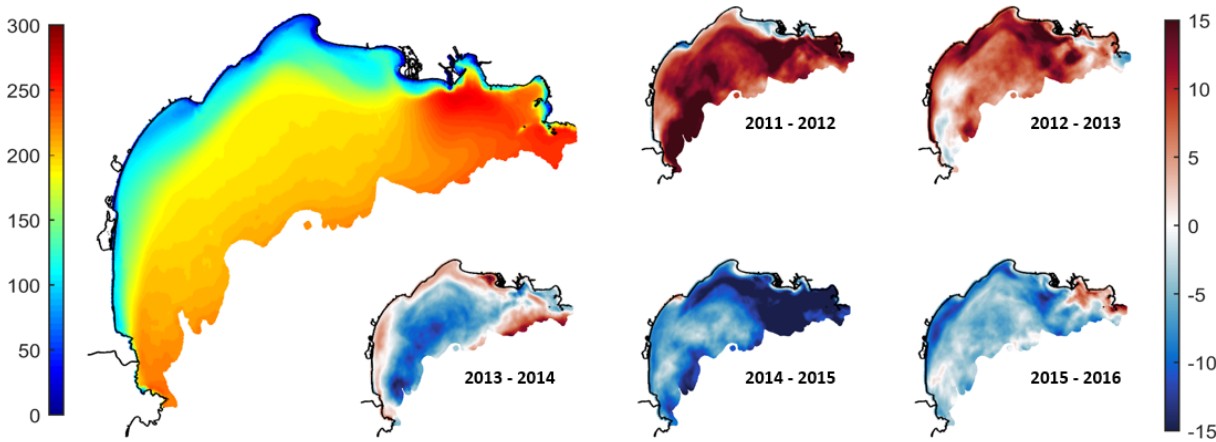

**Figure 11.** The left shows vertically integrated net primary production (in $\mathrm{g\,C\,m^{-2}\,yr^{-1}}$) averaged over the 2011–2016 period. The right shows annual anomalies in the vertically integrated NPP (in $\mathrm{g\,C\,m^{-2}\,yr^{-1}}$).

< 60 m depth) or the bottom layer (Fig. 14b, < 60 m depth). They also show the opposite functioning between the eastern and western parts of the shelf, and in both layers, as in water exchanges.

In the surface layer (Fig. 14a), the overall net transport (yellow line) was in the range of $0$–$0.5 \times 10^5$ t C per month oriented towards the open sea. Exports occurred mainly in the eastern part of the shelf. They reached maximum values in spring ($1.1 \times 10^5$ t C per month during May 2013) with low values in winter and moderate values in summer ($\sim 0.3 \times 10^5$ t C per month). The imports occurred mainly at the end of spring and in summer in the west. They were at a minimum in winter and occurred episodically in autumn in the eastern part of the shelf (maximum in November 2011).

In the bottom layer (Fig. 14b), the overall net transport was lower than in the surface layer, with residual export in autumn and winter and episodic imports without seasonality. Exports occurred mainly in the western part of the shelf. They reached maximum values in the winters of 2012, 2013, and 2015 (maximum of $\sim 0.4 \times 10^5$ t C per month) and also in autumn with slightly lower values. During spring and summer, deep imports of POC in the west occurred, slightly compensated for by deep exports in the east.

## 5  Discussion

A first multi-year assessment with a 3D coupled hydrodynamic–biogeochemical model was presented to quantify POC fluxes on the Gulf of Lion shelf. The model reproduces the annual cycle of nutrient and phytoplankton concentrations in the Gulf of Lion shelf well. The coupling of hydrodynamic–biogeochemical models highlights the role of physical processes such as stratification and winter mixing, which impact nutrient dynamics in the upper layer in agreement with previous observational (Diaz et al., 2000) and modelling studies (Tusseau-Vuillemin et al., 1998;

Herrmann et al., 2013) in the NW Mediterranean region. Dynamics in phytoplankton biomass are then well represented through the spring bloom, the summer DCM, and the erosion of the DCM in autumn, related to the stratification and nutricline dynamics.

### 5.1  Annual cycle of nutrients and physical forcings

We have highlighted the year-round import of nutrients on the shelf from offshore waters of about $22.8 \times 10^4$ t N yr$^{-1}$ for nitrate and $2.9 \times 10^4$ t P yr$^{-1}$ for phosphate that represent 3 and 15 times more, respectively, than annual Rhône inputs ($7 \times 10^4$ t N yr$^{-1}$–$0.2 \times 10^4$ t P yr$^{-1}$), the difference between nitrate and phosphate being explained by the very high N : P ratio in Rhône River inputs (approx. 80 CE2) in agreement with previous studies (Pujo-Pay et al., 2006; Ludwig et al., 2010; Auger et al., 2011). Minimal values estimated in summer can be attributed to weak water exchanges in the deeper layer in this season (Fig. 7). In autumn, exchanges increase (> 20 000; 2500 t P per month) as water exchanges take place on a thicker layer including the nutricline (Figs. 7–9). Most often these events correspond to autumn marine storms. It seems that they can also be partly attributed to the rapid increase in the transport of the Northern Current in November, which corresponds to the activation of its east Corsica branch (Carret et al., 2019). This increase would favour intrusions of the Northern Current on the shelf. Autumn is the beginning of the period also including winter during which the nutrient inventories increase by a factor of 2 to 3 partly due to the low consumption associated with the low solar radiation. In addition, the first two winters were also marked by significant net imports of nutrients on the shelf from February to April 2012 and March 2013. Winter 2012 was marked by extremely dense shelf water formations followed by intense cascading in the canyon of Cap de Creus (Durrieu de Madron et al., 2013). However, as shown in Figs. 7–8, the export of nutrients through dense shelf water cascading in the south-

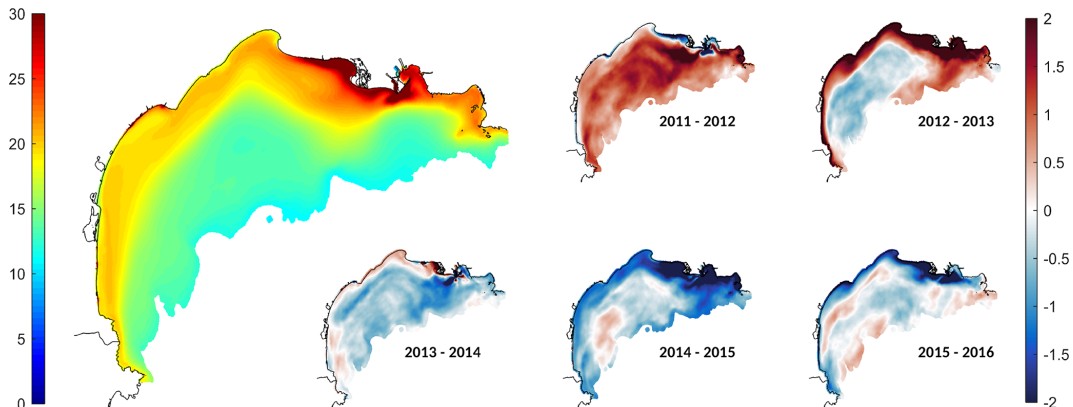

**Figure 12.** The left shows the mean POC deposition over the shelf (in $\mathrm{g\,C\,m^{-2}\,yr^{-1}}$) for the 2011–2016 period. The right shows annual anomalies in the POC deposition (in $\mathrm{g\,C\,m^{-2}\,yr^{-1}}$).

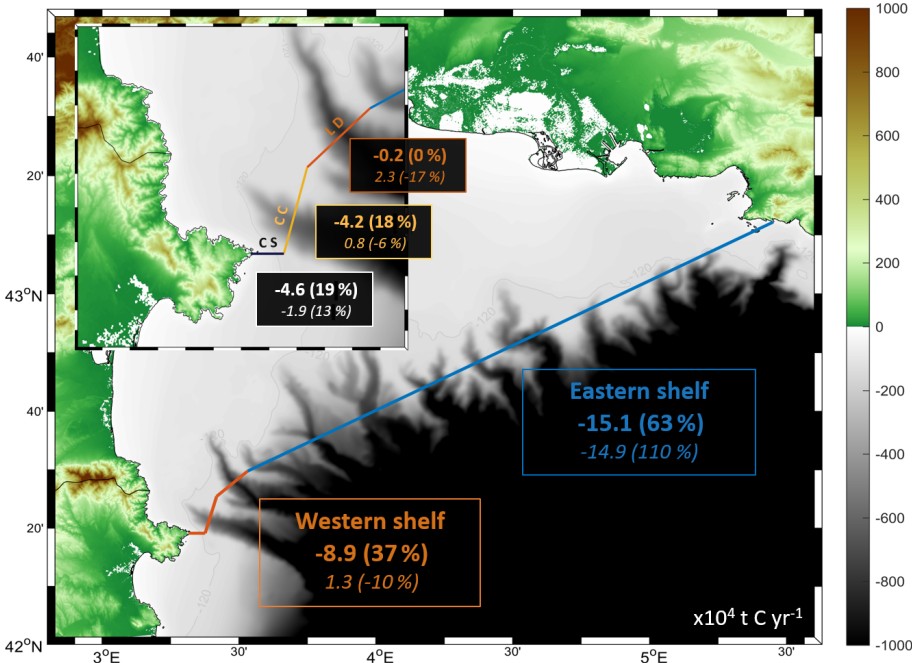

**Figure 13.** Map of the Gulf of Lion's topography (m) showing the position of the different sections used to describe the POC net transport ($10^4\,\mathrm{t\,C\,yr^{-1}}$) in the vicinity of the Gulf of Lion slope and at the eastern and western shelf extremities. By convention net transport off the shelf is shown by a negative value. The detail of the western shelf net transport is shown in the top left corner of the map. Note that here the western section is separated into three subsections. The bold numbers show the net transport vertically integrated over the water column. The numbers in italics show the surface POC net transport ($< 60\,\mathrm{m}$ depth).

western region was exceeded by nutrient inputs in the eastern part of the shelf where about $30\,000\,\mathrm{t\,N}$ per month was advected for 3 months (February–April 2012). In March 2013, the strong nutrient input corresponded to the interaction between offshore deep convection and a southeasterly storm. First, intense vertical mixing produced a nutrient enrichment in the euphotic layer (nitrate concentration up to $8\,\mathrm{\mu mol\,L^{-1}}$ near the surface vs. $3\,\mathrm{\mu mol\,L^{-1}}$ over the shelf; see Kessouri et al., 2017) in the deep convection area from mid-January to early March. Then, when the vertical mixing stopped, a

strong easterly storm advected the surface-nutrient-enriched open sea water onto the shelf. The cyclonic circulation induced by the southeasterly storm on the shelf (Ulses et al., 2008) favoured the import of nutrients in the eastern parts through the surface and deep layers and an export of nutrients in the western parts, mainly in the deep layers. This led to a large net input of nutrients that were partly consumed in the surface layer by phytoplankton. Our results are consistent with the results of the modelling study by Tusseau-Vuillemin et al. (1998), who showed such export through dense wa-

ter cascading, as well as shelf enrichment with nutrients advected from the open sea convection area.

Throughout the year, it is remarkable that all the physical processes described here, whether they correspond to water input from the eastern or western part of the shelf, at the surface, or in the deep layer, result in a net import of nutrients to the Gulf of Lion shelf. This reveals a systematic consumption of nutrients during the transit of the water masses on the shelf, with the water leaving the shelf being poorer than the incoming water. The Gulf of Lion shelf is therefore generally a reactor that consumes the nutrients imported from the open sea year-round to produce planktonic biomass. In more detail, since nutrients are imported mainly under the nutricline, they probably play a major role in stratified periods (summer and autumn) sustaining the production in the DCM and feeding upwelling at the coast. In contrast, in winter, despite high nutrient inputs from the open ocean, the ecosystem is less productive due to low solar radiation and vertical mixing that further reduce the exposure of cells to light. During this season, the dynamics of the shelf linked to the strong winds, combining vertical mixing and Ekman transport towards the open sea, leads to a vertical circulation of nutrients that enter through the bottom and exit through the surface layer, generally with little benefit for the ecosystem.

The primary production is thus impacted by the nutrient availability in the photic layer imported not only from local bottom waters by vertical mixing or by the rivers but also and importantly from offshore waters, particularly from the Northern Current and even from the deep convection region through marine-storm-induced circulations (Conan et al., 1998; Tusseau-Vuillemin et al., 1998).

## 5.2 Biological production

Considering a shelf area of approx. 10 000 km$^2$, we estimate an averaged NPP of 196 g C m$^{-2}$ yr$^{-1}$. This result is in line with previous studies in the NW Mediterranean that estimated an annual NPP in the range of 80–150 g C m$^{-2}$ yr$^{-1}$ using local and punctual in situ measurements (Cruzado et Velasquez, 1990; Lefevre et al., 1997; Conan et al., 1998; Durrieu de Madron et al., 2000), or the range of 160–300 g C m$^{-2}$ yr$^{-1}$ from remote sensing (Bosc et al., 2004; Olita et al., 2011) or of 75–250 g C m$^{-2}$ yr$^{-1}$ using biogeochemical models (Lazzari et al., 2012; Teruzzi et al., 2018). Our estimates are also in the upper range of NPP observed over shelves in mid-latitude areas such as in the Bering Sea, the North Sea, and the Mid-Atlantic Bight (100–150 g C m$^{-2}$ yr$^{-1}$; see details in Hofmann et al., 2011). The estimates of the annual NPP obtained on the shelf are slightly higher than the ones estimated with the same biogeochemical model in the deep sea convection ranging between 150 and 175 g C m$^{-2}$ yr$^{-1}$ (Ulses et al., 2016; Kessouri et al., 2018), also showing weak interannual variability.

The high productivity and the recycling of organic matter during the year were highlighted. Positive NEP values

year-round (max. NEP of $\sim$ 5000 t C d$^{-1}$ during April 2015) show that the GoL shelf acted as a sink of DIC (source of organic carbon) regarding the pelagic planktonic ecosystem (i.e. the biological term). In addition, the NEP decreased during autumn and was minimum in winter due to the decrease in the GPP while the total respiration recycled the OC. Negative daily NEP values (not shown) occurred in December 2012 and 2014 and in November 2013, which shows that the GoL shelf ecosystem episodically acted as a source of DIC (Sempéré et al., 2000). Our annual estimate of $90.6 \pm 3.3 \times 10^4$ t C yr$^{-1}$ is 43 % higher than the estimate of Sempéré et al. (2000).

Primary production (NPP) at the annual scale shows a weak interannual variation with a standard deviation of 4 %. It shows a strong spatial variability with a range from 50 g C m$^{-2}$ yr$^{-1}$ in the very shallow regions to 200 g C m$^{-2}$ yr$^{-1}$ on the outer shelf and 250 g C m$^{-2}$ yr$^{-1}$ in the eastern part of the shelf. The main spatial variability with a coast–open sea gradient is caused by the increasing reservoir of available nutrients with depth. This spatial pattern is consistent with the study of Macias et al. (2018) showing maximal primary productivity in the deeper and eastern regions of the shelf. Along with this general pattern, changes in environmental variables are expected to play a role in the changes in the NPP over the shelf. Hence, we aim at identifying the key biogeochemical, hydrodynamic, and atmospheric indicators, which could potentially explain the additional spatiotemporal variability in NPP over the simulated period. To that end, we determined the empirical orthogonal function (EOF) decomposition and the major principal components (PCs) of NPP weekly anomalies (Fig. 15) (see details in Olita et al., 2011, and Daewel and Schrum, 2017).

The first EOF explains 64 % of the NPP variability with positive values over the entire shelf and higher values in the eastern region and along the continental slope. Its temporal variability (PC$_1$) follows the pattern of the temporal variability of the solar radiation with a correlation of 0.43 ($p < 0.001$) (Fig. 15). In addition, the second EOF explains 14 % of the NPP changes with an opposite behaviour between the coastal area ($< 50$ m deep, mainly in the Bay of Marseille and the SW part of the GoL) and the central part of the shelf (see Fig. 15). Its temporal variability (PC$_2$) is significantly correlated with the weekly anomalies of wind intensity. At last, the third EOF explains 9 % of the NPP changes with the opposite behaviour between the Rhône ROFI and the rest of the shelf. It is related to the temporal variability of the Rhône river NO$_3$ anomalies (PC$_3$; see Fig. 15 – bottom panel; same results are observed with PO$_4$ anomalies). PCs are thus related to processes that drive the nutrient concentrations, controlling the primary production.

– *EOF$_1$/PC$_1$* CE3. Changes in NPP over the entire shelf are positively correlated to solar radiation. This indicates that the primary production over the shelf is controlled by the nutrients available (see Sect. 5.1) and regulated

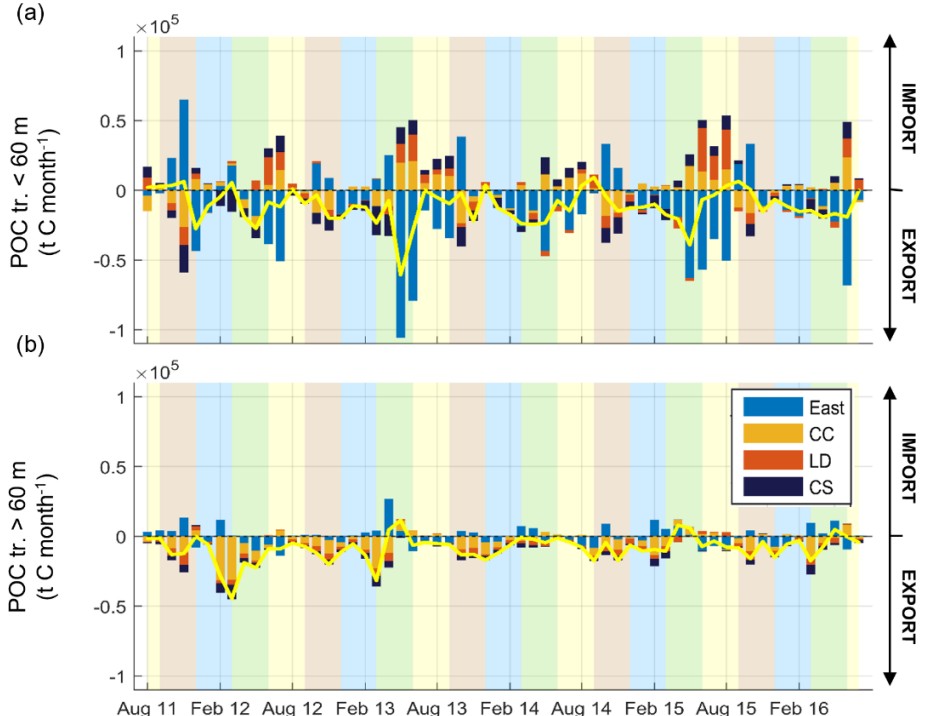

**Figure 14. (a)** Temporal variability of the monthly net surface ($< 60\,\mathrm{m}$) transport of POC (t C per month) through the sections defined in Fig. 13. **(b)** Same as **(a)** but for depths superior to 60 m. Note that here the western section is separated into three subsections. By convention, import (export) of POC is shown by positive (negative) values. The residual net transport is shown by the yellow line. Shaded areas show the different seasons over the period simulated (yellow, JJA; brown, SON; blue, DJF; green, MAM).

by the anomalies of solar radiation (i.e. light intensity). Maximum production periods thus occur when inputs of nutrients are high (from offshore waters or Rhône River) and are concomitant with periods of strong solar radia­tion positive anomalies (Legendre, 1990).

– $EOF_2/PC_2$ CE4. Changes in NPP in the bay of Marseille and along the coast of the GoL are positively corre­lated to the wind speed. This indicates the role of wind-induced coastal upwellings in the supply of nutrients to the surface layer that locally favours the primary production (Lefevre et al., 1997; Fraysse et al., 2014) and wind-induced eastern storms when the nutrient-rich Rhône plume is pushed towards the coast and flows southwestward (Ulses et al., 2008a) and may in certain cases be diluted towards the east (Gatti et al., 2006).

– $EOF_3/PC_3$ CE5. Changes in NPP in the Rhône ROFI are positively correlated with the Rhône nutrient input anomalies (same results with $PO_4$, not shown). It shows the role of the Rhône floods in the supply of nutrients to the gulf, which locally increases the NPP (Minas and Minas, 1989; Durrieu de Madron et al., 2003).

## 5.3 The deposition and offshore export of POC

In terms of POC deposition (Fig. 12), the model reproduces the high accumulation rates observed in front of the Rhône mouth (Cathalot et al., 2013). In addition, the cyclonic cir­culation favours the alongshore dispersion of the terrestrial material from the Rhône River along the 30–50 m isobaths (Got and Aloisi, 1990; Durrieu de Madron et al., 2000). An­nual changes can be related to changes in Rhône terrestrial POC inputs to the shelf (minimum inputs in 2014–2015 and 2015–2016 and maximum inputs in 2012–2013 correspond­ing to minimum and maximum deposition in front of the Rhône mouth, respectively) as well as to the inter-annual variability of the NPP (see above) that drove the deposition of POC over the shelf (high NPP in 2011–2012 and 2012–2013) in agreement with Auger et al. (2011). In addition, it is also noteworthy that these results do not take into account possible wave-induced sediment resuspension processes dur­ing storms, which participate in changes in sediment depo­sition areas (Bourrin et al., 2015). The future development of a fully coupled hydrodynamic, sedimentary, and biogeo­chemical model could provide a better description of the dy­namics of POC deposition, in particular on the inner shelf (0–30 m). Among the deposited sediments, the average rem­ineralization rate of POC was estimated to be 70 %, leading to a loss by degradation of about $13.8 \pm 1.3 \times 10^4\,\mathrm{t\,C\,yr}^{-1}$

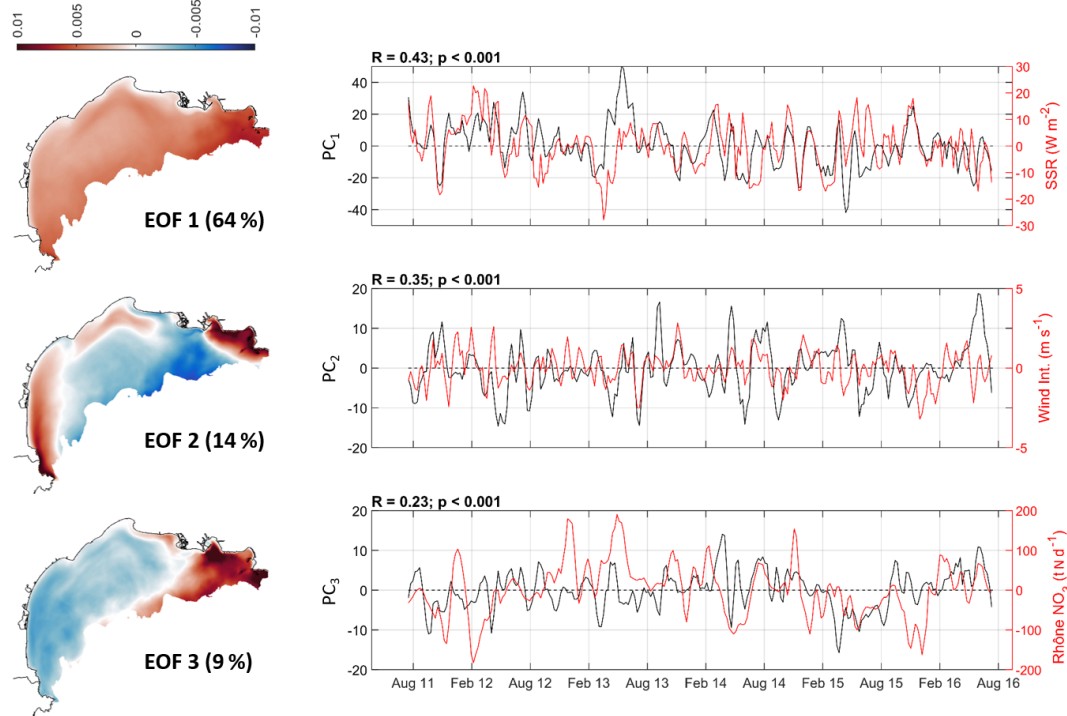

**Figure 15.** The left shows the first three empirical orthogonal functions (EOF) for the weekly anomalies of NPP in the Gulf of Lion. The right shows the principal components for the pattern of the first three EOFs (black). Each PC is related to the weekly anomalies of the related indicator (red), from top to bottom: the solar radiation, the wind intensity, and the Rhône $NO_3$ input. Weekly anomalies are estimated by subtracting the average annual cycle for each simulated year. A 3-week smoothing filter is then applied.

in the sediment (Table 2). This result is in line with Accornero et al. (2003) and Pastor et al. (2011), who estimated a mean remineralization rate of 60 % in the Gulf of Lion, and with Durrieu de Madron et al. (2000), who estimated a total loss by the degradation of $33.8 \pm 16.1 \times 10^4 \, \text{t C yr}^{-1}$ at the water–sediment interface.

Our results have highlighted the first order of importance of the cyclonic circulation that occurs over the shelf and that favours water and POC export in the southwestern part of the shelf ($\sim 2900 \, \text{t C yr}^{-1} \, \text{km}^{-1}$ across the western boundary compared to $\sim 850 \, \text{t C yr}^{-1} \, \text{km}^{-1}$ for the eastern one). They agree with the results from previous studies, which showed the important water and particulate matter export in this part of the shelf due to the winter coastal circulation (Estournel et al., 2003; Ulses et al., 2008c) and episodic marine storm events (Ulses et al., 2008a, b; Bourrin et al., 2015). Lapouyade and Durrieu de Madron (2001) estimated from in situ measurements a mean transport of $3.3 \times 10^4 \, \text{t C}$ per month ($12.8 \, \text{kg s}^{-1}$) in winter that drastically decreased to $0.2 \times 10^4 \, \text{t C}$ per month ($0.8 \, \text{kg s}^{-1}$) during the summer. These estimates are of the same order as those obtained here, with mean winter and summer POC transport of $2.3 \times 10^4$ and $0.5 \times 10^4 \, \text{t C}$ per month.

The estimates of POC transport for individual parts of the western slope (see Fig. 13) demonstrate the importance of the Cap de Creus Canyon area and the adjacent narrow part of the shelf (CS in Fig. 13) in the export of POC off the GoL (4.2 and $4.6 \times 10^4 \, \text{t C yr}^{-1}$, respectively). It is also remarkable that this transport is maximum in winters 2011–2012 and 2012–2013, during a period marked by several cascading events and marine storms (Durrieu de Madron et al., 2013; Bourrin et al., 2015; Mikolajczak et al., 2020). This is consistent with the analyses on the interannual variability of shelf water export in the southwestern region by Mikolajczak et al. (2020) based on winter meteorological conditions simulated by ECMWF over the period 2009–2017. Over our study period, they identified two mild winters, 2013–2014 and 2015–2016, and two cold winters, 2011–2012 and 2012–2013, favourable to shelf water formation and cascading, as well as to deep export in the southwestern canyons. The winter of 2014–2015 also appeared as a cold winter in their analysis but observations in the Lacaze-Duthiers canyon at 500 and 1000 CE6 depth indicate no signal of dense water cascading for this winter, suggesting a mitigation of shelf dense water cascading by other processes. Their analysis also showed that the winter of 2012–2013 was also marked by a higher-than-average number of days of strong easterly winds (i.e. 7) contrary to the other winters studied in this work. The southwestern part of the shelf is an active area in the export of POC towards the open sea and the deeper environ-

**https://doi.org/10.5194/bg-18-1-2021** **Biogeosciences, 18, 1–26, 2021**

Please note the remarks at the end of the manuscript.

ments as highlighted in Puig et al. (2008) and Sanchez-Vidal et al. (2008). In addition, it is noticeable that the Lacaze-Duthiers canyon area shows a balanced net transport of POC with maximum imports in summer of 2013 and 2015 and low exports in winters. This probably highlights the balance existing between the inputs from the open sea during the stratified period and continental winds that generate anticyclonic eddies in this area (Estournel et al., 2003; Hu et al., 2011) and outputs from transport during extreme events such as storms and dense shelf water cascading (Palanques et al., 2006).

We estimate that approx. $4.6 \pm 2.8 \times 10^4\,\mathrm{t\,C\,yr^{-1}}$ was exported towards the Catalan margin and the Gulf of Rosas, by-passing the Cap de Creus Canyon. This value, which represents approx. 20 % of the total net transport off the shelf, shows a significant contribution of POC from the GoL to this area. Future studies could focus on the balance between POC inputs from the southwestern part of the GoL and from local rivers (such as the Tordera, Ter, and Fluvià rivers), estimated at $0.13 \times 10^4\,\mathrm{t\,C\,yr^{-1}}$ by Sanchez-Vidal et al. (2013). Observations in this part of the shelf could also validate our estimates in this area, which has been seldom instrumented from now.

Considering surface waters, the impact of the steady northwestern wind that favours offshore water exports through eddies and intrusions of the Northern Current has been identified as the main factor regulating surface shelf–slope exchanges in this area (Estournel et al., 2003; Petrenko, 2003). The surface export of POC is also related to the spread of the Rhône River plume towards the open sea during floods and surface current favourable conditions (Gangloff et al., 2019; Many et al., 2018). The high concentrations of POC in the plume therefore actively contribute to offshore exports through the Northern Current during these episodic events.

Finally, considering the continental shelf area ($11\,000\,\mathrm{km^2}$), our estimate of the net cross-shelf export corresponds to a mean value of $21.7\,\mathrm{g\,C\,m^{-2}\,yr^{-1}}$. As a matter of comparison, the export of POC below the euphotic zone in the northwestern deep convection area characterized by intense winter mixing was estimated in previous modelling studies to be between 15 and $30\,\mathrm{g\,C\,m^{-2}\,yr^{-1}}$ at $100\,\mathrm{m}$ by Guyennon et al. (2015), to be $27.4 \pm 6.6\,\mathrm{g\,C\,m^{-2}\,yr^{-1}}$ at $100\,\mathrm{m}$ depth by Ulses et al. (2016), and to be $25\,\mathrm{g\,C\,m^{-2}\,yr^{-1}}$ at $150\,\mathrm{m}$ by Kessouri et al. (2018). The modelling results of Ulses et al. (2016) showed that the surface layer (surface to $100\,\mathrm{m}$ depth) of the northwestern Mediterranean open sea was a sink of organic carbon for surrounding shallow areas and a net source of organic carbon for the western and southern open sea of the western Mediterranean Sea. The POC GoL shelf could thus partly transit by the northwestern open sea before being transferred to the rest of the western and southern open seas in the surface layer or in the deeper layer after downward export.

## 6 Conclusion and future works

A 3D coupled hydrodynamic–biogeochemical model was used to estimate the POC dynamics in the Gulf of Lion shelf over the period 2011–2016. The validation of the model was performed using existing data from a multi-platform observation system. A good agreement between model results and observations at different space scales and timescales was shown. The model represents the high dynamical character of the Gulf of Lion shelf in terms of hydrodynamic (stratification and winter mixing) and biogeochemical conditions (nutrients, chlorophyll $a$, and POC dynamics).

Spatial, seasonal, and inter-annual changes in the different POC input and output terms were identified. Model results highlight the high NPP occurring in the GoL shelf ($198.9 \times 10^4\,\mathrm{t\,C\,yr^{-1}}$), with maximum values in spring and in the outer shelf, in particular in the eastern region. The interannual variability of the NPP at the Gulf of Lion scale is especially low (SD $= 4\,\%$), and monthly interannual anomalies in NPP are mainly explained by changes in the intensity of solar radiation. Our results also show that the nutrient enrichment from the general circulation and the open sea ($22 \times 10^4\,\mathrm{t\,N\,yr^{-1}}$ and $3 \times 10^4\,\mathrm{t\,P\,yr^{-1}}$), representing on average 3 times the inputs from the Rhône for nitrate and 15 times the inputs from the Rhône for phosphate, favours the phytoplankton growth over the entire shelf. Coastal upwelling and inputs for the Rhône also contribute locally and to a lesser extent to changes in NPP. The positive NEP values during a large part of the year show that the GoL annually acts as a sink of DIC regarding the pelagic planktonic ecosystem.

Rivers contribute to the POC delivery to the shelf with a mean value of $19 \times 10^4\,\mathrm{t\,C\,yr^{-1}}$, representing 10 % of the NPP. At last, we have shown the high dynamical character of the GoL shelf waters, which results in strong cross-shelf transport of POC oriented off-shelf ($24 \times 10^4\,\mathrm{t\,C\,yr^{-1}}$). Our estimates have shown that 57 % of the export occurred in the surface layer through the eastern and central parts of the shelf, and 43 % occurred below $60\,\mathrm{m}$ mostly through the western part of the shelf and favoured by marine winds in autumn and cascading events in winter.

At the scale of the western Mediterranean Sea, our results show the crucial role of the Gulf of Lion shelf, which acts as a reactor that consumes the inorganic nutrients imported from the open sea to produce organic matter year-round. A part of the produced organic matter is exported by the coastal circulation towards the Catalan shelf and the open sea, yielding the enrichment in POC of adjacent coastal waters, the Northern Current, and the deeper basin. The autotrophic status of the ecosystem along with the highly dynamical character of the area, marked by low residence times and shelf–slope processes as dense shelf water cascading, suggest that the GoL shelf could act as a sink for atmospheric $CO_2$ and favour carbon sequestration. The future coupling of the model presented in this work with a carbonate system module describ-

ing the dynamics of dissolved inorganic carbon and estimating the air–sea $CO_2$ fluxes could allow a better understanding and a quantification of the source–sink role of atmospheric $CO_2$ on the shelf, as well as an estimate of a closed budget of carbon for the area.

This work represents a first step for further investigations. Along with the coupling with a module describing the dynamics of the carbonate system, a coupling with a module of sediment transport would allow a better representation of the tight link between pelagic and benthic processes occurring in shallow regions, notably the fluxes of organic carbon and inorganic nutrients contained within the sediment, taking place at the sediment–water interface during storm-induced resuspension. These events are likely to significantly impact primary production as well as the rate of sedimentation and benthic remineralization on the wave-influenced inner shelf. Moreover, the northwestern Mediterranean Sea is subject to multiple global and regional stressors. The ongoing sea warming and increasing stratification of the water column (Somot et al., 2006; Darmaraki et al., 2019) could affect the metabolism rates and ecosystem dynamics. The results of Herrmann et al. (2014), based on the same biogeochemical model applied to annual future periods (end of the 21st century) under the SRES-A2 climate change scenario, showed that the water warming could induce an increase in primary production, respiration, DOM CE7 exudation, and bacterial growth rates at the scale of the northwestern Mediterranean Sea. In their scenarios, the net metabolism would not vary significantly, and the microbial activity would be enhanced. The future projections performed at the scale of the whole Mediterranean Sea by Lazzari et al. (2014), under SRES-A1B scenarios and different river load conditions for a 20-year period, as well as the transient projections by Moullec et al. (2019) under the RCP8.5 scenario also predicted an increase in rates of primary production and respiration at the end of the 21st century, due to the increase in sea temperature. Lazzari et al. (2014) found a change in the distribution of organic carbon with a decrease in the living organism particulate biomass and an increase in the dissolved organic carbon inventory. This resulted from a reduction of nutrient availability in response to enhanced stratification. However, based on transient simulations over the entire Mediterranean Sea, the future projections of Richon et al. (2019) (under SRES-A2 scenario) and Pagès et al. (2020) (under a RCP8.5 scenarios) showed an opposite trend, with, in both studies, a decrease in primary production on the shelf of the Gulf of Lion due to a depletion in nutrients (phosphate and nitrate, respectively), by the end of the 21st century. The variability of the response of primary production to climate change in the various modelling studies partly lies in the temperature sensitivity in model equations of primary production as pointed out by Taucher and Oschlies (2011). Indeed the rate of primary production in the model used by Herrmann et al. (2014), Lazzari et al. (2014), and Moullec et al. (2019) is dependent on temperature through an exponential function (Eppley et al., 1972; Eq. S58). Finally, the increasing stratification could also affect the shelf–slope exchange processes, impacting the transport of organic carbon from the Gulf of Lion shelf toward the deeper basin. Herrmann et al. (2008) showed a reduction of 90 % of shelf dense water cascading on the Gulf of Lion shelf at the end of the 21st century based on the A2 climate change scenario. This suggests that the transfer of shelf carbon to the deep sea could be drastically reduced and mostly "re-directed" towards the Catalan shelf. In future works, we plan to contribute to the investigation of the impact of climate change on the biogeochemical, physical, and air–sea carbon fluxes by developing high-resolution modelling in this shallow area characterized by complex processes at their various interfaces (sea–air, sediment–water, and land–sea).

*Code availability.* . TS15

*Data availability.* . TS16

*Supplement.* The supplement related to this article is available online at: https://doi.org/10.5194/bg-18-1-2021-supplement.

*Author contributions.* . TS17 .

*Competing interests.* The authors declare that they have no conflict of interest. TS18 .

*Acknowledgements.* Numerical simulations were performed using the SYMPHONIE model, developed by the SIROCCO group (https://sirocco.obs-mip.fr/ TS19 ) and computed on HPC resources of CALMIP (CALcul en MIdi-Pyrénées, projects 1331, 1325 and 09115) and GENCI (Grand Equipement National de Calcul Intensif, project A0010110088). Datasets were retrieved from the SOMLIT network (Service d'Observation en Milieu Littoral; http://somlit.epoc.u-bordeaux1.fr/fr TS20 ). We thank SOERE MOOSE for supporting and providing long-term observation data in the Gulf of Lion. We especially thank the French glider team, P. Testor, and F. Bourrin TS21 for their help in the glider data processing.

*Financial support.* This work was funded by the ANR AMORAD project under the ANR programme (ANR-11-RSNR-0002). The postdoctoral fellowship of Gaël Many was also funded by ANR AMORAD. This study is also a contribution to the DeltaRhone

project funded by the EC2CO (Ecosphere Continentale et Cotiere) programme. TS22

*Review statement.* This paper was edited by Yuan Shen and reviewed by two anonymous referees.

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

### Remarks from the language copy-editor

CE1    Please note that this manuscript has undergone copy-editing according to the standards of Oxford English.

CE2    Should this value have a unit? Please check and advise.

CE3    Is this a ratio? Or do you mean $EOF_1$ and $PC_1$? Please clarify.

CE4    Is this a ratio? Or do you mean $EOF_2$ and $PC_2$? Please clarify.

CE5    Is this a ratio? Or do you mean $EOF_3$ and $PC_3$? Please clarify.

CE6    Should these values have the unit "m" for "metres"? Please check and advise.

CE7    Please define.

### Remarks from the typesetter

TS1    The composition of Figs. 2, 6–10 and 13–15 has been adjusted to our standards.

TS2    Please provide running title.

TS3    Please add last access date.

TS4    Please add last access date.

TS5    Here (and potentially elsewhere in the manuscript) you are referring to data and/or data sets. Please see our remark in the data availability section regarding our standards.

TS6    Please add last access date.

TS7    Please add last access date.

TS8    Please check link.

TS9    Please add last access date.

TS10    Please add last access date.

TS11    Please confirm.

TS12    Please confirm.

TS13    Please confirm.

TS14    Please check.

TS15    Please provide a statement on how your underlying software code can be accessed. If the code is not publicly accessible, a detailed explanation of why this is the case is required. The best way to provide access to software code is by depositing it (as well as related metadata) in reliable public repositories, assigning digital object identifiers (DOIs), and properly citing code as individual contribution. Please indicate if different software codes are deposited in different repositories or if code from a third party was used. Additionally, please provide a reference list entry including creators, title, and date of last access. If no DOI is available, assets can be linked through persistent URLs to the software code itself (not to the repositories' home page). This is not seen as best practice and the persistence of the URL must be secured.

TS16    You have referred to data sets in your text. Please provide a statement on how these underlying research data can be accessed. If the data are not publicly accessible, a detailed explanation of why this is the case is required. The best way to provide access to data is by depositing them (as well as related metadata) in reliable public data repositories, assigning digital object identifiers (DOIs), and properly citing data sets as individual contributions. Please indicate if different data sets are deposited in different repositories or if data from a third party were used. Additionally, please provide a reference list entry including creators, title, and date of last access. If no DOI is available, assets can be linked through persistent URLs to the data set itself (not to the repositories' home page). This is not seen as best practice and the persistence of the URL must be secured.

TS17    Please note that the section "Author contributions" is mandatory. Please provide the text for this section in complete sentences. Please see https://publications.copernicus.org/for_authors/obligations_for_authors.html for more information

TS18    Declaration of all potential conflicts of interest is required by us as this is an integral aspect of a transparent record of scientific work. If there are possible conflicts of interest, please state what competing interests are relevant to your work. Please see https://publications.copernicus.org/services/competing_interests_policy.html

TS19    Please add last access date.

TS20    Please add last access date.

TS21    Please provide full author names.

TS22    Please note that there is a discrepancy between funding information provided by you in the acknowledgements and the funding information you indicated during manuscript registration, which we used to create this section. Please double-check your acknowledgements to see whether repeated information can be removed from the acknowledgement or changed accordingly. If further funders should be added to this section, please provide the funder names and the grant numbers. Thanks.

TS23    Please ensure that any data sets and software codes used in this work are properly cited in the text and included in this reference list. Thereby, please keep our reference style in mind, including creators, titles, publisher/repository, persistent identifier, and publication year. Regarding the publisher/repository, please add "[data set]" or "[code]" to the entry (e.g. Zenodo [code]).

TS24    Please add page range or article number.

TS25    Please name all authors.

TS26    Please add journal.

TS27    Please add page range or article number.

TS28    Please add page range or article number.

TS29    Please add page range or article number.

TS30    Please name all authors.

TS31    Please add page range or article number.

TS32    Please add page range or article number.

TS33    Please add page range or article number.

TS34    Please add page range or article number.

TS35    Please add page range or article number with DOI.

TS36    Please add page range or article number.

TS37    Please add page range or article number.

TS38    Please add page range or article number.