# Peer review of "Particulate organic carbon dynamics in the Gulf of Lion shelf (NW Mediterranean) using a coupled hydrodynamic-biogeochemical model"

_Biogeosciences, 2021_

## Author Response (AR1)

**Response to reviewer 1**

We thank you very much for your constructive and relevant comments and suggestions. Below the reviews are reproduced in black font and our responses interspersed in blue and updates of the text in green.

The study by Many et al., used the 3-D numerical model to simulate the physical and biogeochemical processes in the Gulf of Lion shelf, one of the well-characterized coastal regions in the world ocean. The encouraging agreement between model projections and field measurements gives confidence in the accuracy and rationality of model simulations. With the particular focus on the POC budget, temporal and spatial variability of multiple POC fluxes and associated underlying mechanisms were discussed. Overall, the manuscript is well organized and this work represents an important step toward better understanding the interactions between physical and biogeochemical processes as well as the regional carbon cycling. However, some major concerns need to be addressed before getting published.

Reply: We appreciate this overall positive assessment and believe we can address the Reviewer 1 concerns as detailed below.

Major concerns:

1) Confusion on the research goal: toward closing regional POC budget or just analyzing the spatiotemporal variability of some of key POC fluxes? "POC budget" was mentioned in the title and many places throughout the main text. In principle, the "budget" means the effort to balance the time rate change of POC inventory by the multiple processes including biological activity and physical transport. If so, the paper should start with the introduction about the mass balance equation (i.e. $POC/dT = POC_{bio} + POC_{horizotnal}$ advection + $POC_{depostion} + POC_{export} + ....$) and go over the main processes. From the mass balance perspective, the NEP is the best term to represent the net biological process in governing the time rate change of POC and partition into GPP, NPP CR and seems redundant. Also, given that horizontal advection is important as the author mentioned in the introduction, it should be discussed in the main text. I envision the paper should end with a schematic diagram, something like a box showing how different processes balance the change of POC in the seawater. However, in the current version, the authors seem to focus on some POC fluxes that authors are interested in rather than a comprehensive overview of POC fluxes with the aim to balance the time rate change of POC. I am not saying the present way is wrong. I am open to both strategies and it depends on the study goal. Therefore, I think the author should be cautious in using "POC budget" and be more clear about the research goal.

Reply: We agree with the comment of the reviewer. As we plan to close the carbon budget in a future work (i.e. with the integration of the inorganic carbon cycle in the model), we here focus on the POC dynamics through the study of the main POC fluxes over the shelf. The manuscript title and main text will be adapted toward this goal. We will thus change all the mentions of "budgets" to "dynamics" in the revised version, notably in the title.

2) Issue about DOC portion in GPP, NPP and respiration: the author refers GPP, NPP and respiration to one of POC fluxes. Primary production and respiration both include POC and DOC production, even though some field measurements of primary production (i.e. $^{14}$C-based approach) is biased toward POC production because of methodological problems. I am not mistaken, primary production and respiration in the model encompass both DOC and POC portions. In the coastal region, the DOC production/consumption are significant. Since this study focuses on POC dynamics, did the author pay any effort to isolate the DOC portion in these biological terms?

Reply: The biogeochemical model ECO3M-S simulates the biogeochemical cycles of C, N, P, O$_2$ and the dynamics of the main nutrients in the Mediterranean Sea, NO$_3$, NH$_4$, PO$_4$, SiO$_4$, 3 size-classes of phytoplankton, 3 size-classes of zooplankton, one bacteria compartment, dissolved organic matter, light and heavy sinking detrical particles, and dissolved oxygen. Thus, DOC and POC fluxes are calculated separately in our model. To clarify how POC and DOC fluxes were calculated in the model we will add a brief description of the biogeochemical model in Section 2.1.2 and a more detailed description (Text S1) in a Supplementary Material document, with a figure showing the biogeochemical model structure and the biogeochemical processes interacting between compartments (Figure S1) and tables with the list of state variables (Table S1), biogeochemical fluxes and functions (Table S2), parameters (Table S3) and equations of the biogeochemical fluxes (Table S4). In the first version of the manuscript we mainly discussed the POC/DIC fluxes. However we agree with the Reviewer 1 that the magnitude of DOC fluxes in the Gulf of Lion are significant as for instance the DOC exudation (mean of 120 t C yr$^{-1}$) . Therefore to accurately answer this question we will estimate all the POC/DOC fluxes and add their estimates in Section 4.1.3.

3) Missing the information about the methodology in simulating POC fluxes: as the core components, I have not seen the descriptions about how multiple POC fluxes were calculated in the model and definitions about different processes. As mentioned above, how did you calculate the primary production, respiration and partition the POC portion from the total organic carbon term? How did you define/differentiate the POC deposition, cross-shelf transport and horizontal advection? It should introduce in the method section briefly rather than citing the previous paper.

Reply: We hope that the added description of the biogeochemical model in Section 2.1.2 and in Supplementary Material, with the equations of the different fluxes, in the revised manuscript clear up these concerns. The POC deposition is the sum of the concentration of micro-phytoplankton and particulate detritus (in carbon) at the near-bottom level of the model grid, multiplied by their respective settling velocity. To ensure clarity, the term "horizontal advection" will be replaced by cross-shelf transport throughout the revised manuscript. The cross-shelf transport is the flux of water, nutrients or POC through a vertical section along the slope, from the sea surface down to the bottom, shown on Figure 1. A sub-section "2.1.3 Estimation of water, nutrients and POC transport" will be added in Material and Method of the revised manuscript to clarify our methodology.

"Water, nutrients and POC transport are estimated through sections that close off the Gulf of Lion shelf (see Fig. 1). The water column is each time divided into two parts, above and below

60 m corresponding roughly to the depth of the nutricline in summer (Kessouri et al., 2017). The sections are considered down to the bottom with maximum depth depending on the local bathymetry (Figure 1). The "western" section corresponds to the area known to be responsible for deep export by cascading (sometimes down to the bottom of the basin ~2500 m) during cold winters (Ulses et al., 2008c; Durrieu de Madron et al., 2013). This export is restricted to 300-400 m during mild winters and also during eastern storms, which blow predominantly in fall and produce a downwelling in the Cap de Creus Canyon (Ulses et al., 2008a; Mikolajczak et al., 2020). The other section hereafter named "eastern" for the sake of simplicity is known in the eastern part as an intrusion zone of the Northern Current (Conan et al., 1998), while in the center of the shelf, exchanges with the Northern Current have also been (more rarely) documented (Estournel et al., 2003).  It is also the area where the Rhone plume most often exits the shelf under prevailing NW to N wind conditions (Gangloff et al., 2017; Many et al., 2018)."

Minor comments:

Line 170: provide the link for accessing the satellite data.

Reply: The link (https://oceancolor.gsfc.nasa.gov/) to access the satellite data will be added to the manuscript (section 2.2.2).

 Figure 3: add the "surface" and "bottom" on the top of the panel for clarification (like Figure 2).

Reply: Clarification will be made in the Figure.

Figure 6c: Introduce the way to calculate stratification index in Method section.

Reply: Clarification will be added to the manuscript. The definition of the Stratification Index will be added to the section "4.1.1" as it is only used here.

"The stratification index is estimated as the vertical integration of density profiles along depth (expressed in kg m$^{-2}$), then spatially averaged over the shelf. It represents the amount of buoyancy to be extracted to mix the water column from the surface to the bottom and achieve a homogenous density equal to the bottom density."

Table 2: does the "stock POC" mean the POC inventory (t Cyr-1) or the time rate change of POC inventory (t Cyr-1)? The other terms listed in this table are all flux (t Cyr-1).

Reply: Here the stock of POC is the annual mean POC inventory over the shelf, expressed in t C. Fluxes are annually estimated and are thus expressed in t C yr$^{-1}$. This clarification will be explained in the legend of the table. We will change the POC "stock" to POC "inventory" throughout the text.

Section 4.2: Regarding the primary production, do you have a specific reason to focus on NPP rather than GPP or both?

Reply: As the NPP is generally presented in similar modeling studies and measured *in situ*, we focused on its variability to be able to compare and discuss our estimates with previous works (Cruzado et Velasquez, 1990; Lefevre et al., 1997; Conan et al., 1998; Durrieu de Madron et al., 2000; Lazzari et al., 2012).

Line 500: does primary production refer to the NPP or GPP? Please clarify herein

Reply: Here the primary production refers to the NPP. Clarification will be added to the manuscript.

Revise the expressions throughout the text: change Chl-a, umolC L$^{-1}$, NO3 and PO4 to Chl-*a*, umol C L$^{-1}$,  and , respectively.

Reply: Clarification will be added to the manuscript and figures. We will change the "Chl-a" in "Chl-*a*", the "tC, tN, tP" in "t C, t P, t N" (same with µmol, g, and other units).

**References**

Conan, P., Pujo-Pay, M., Raimbault, P., and Leveau, M.: Variabilité hydrologique et biologique du golfe du Lion. II. Productivité sur le bord interne du courant, Oceanologica Acta, 21, 767-782, https://doi.org/10.1016/S0399-1784(99)80005-X, 1998.

Cruzado, A., and Velasquez, Z.R.: Nutrients and phytoplankton in the Gulf of Lions, northwestern Mediterranean, Cont. Shelf Res., 10, 931 – 942, https://doi.org/10.1016/0278-4343(90)90068-W, 1990.

Durrieu de Madron, X., Abassi, A., Heussner, S., Monaco, A., Aloisi, J. C., Radakovitch, O., Giresse, P., Buscail, R., and Kerhervé, P.: Particulate matter and organic carbon budgets for the Gulf of Lions (NW Mediterranean), Oceanologica Acta, 23, 717–730, https://doi.org/10.1016/S0399-1784(00)00119-5, 2000.

Lazzari, P., Solidoro, C., Ibello, V., Salon, S., Teruzzi, A., Béranger, K., and Crise, A.: Seasonal and inter-annual variability of plankton chlorophyll and primary production in the Mediterranean Sea: a modelling approach, Biogeosciences, 9, 217-233, https://doi.org/10.5194/bg-9-217-2012, 2012.

Lefevre, D., Minas, H.J., Minas, M., Robinson, C., Williams, P.J. Le B., and Woodward, E.M.S.: Review of gross community production, primary production, net community production and dark community respiration in the Gulf of Lion, Deep-Sea Research II, 44, 801–832, https://doi.org/10.1016/S0967-0645(96)00091-4, 1997.

**Response to reviewer 2**

We thank you very much for your constructive and relevant comments and suggestions. Below the reviews are reproduced in black font and our responses interspersed in blue and preliminary updates of the text in green.

I think that the manuscript is a useful contribution, as it applies a coupled hydrodynamic – biogeochemical model for the Gulf of Lion (NW Mediterranean) to study the current particulate organic carbon(POC) budget in this continental shelf.

The manuscript is well written and well organized. The model is clearly explained, validated, and the results and discussion are well developed (for the most part; see below). It perfectly fits within the scope of the journal, being of interest to a large group of readers.

Reply: We appreciate this overall positive assessment.

The manuscript is a significant contribution to biogeochemical modeling on continental shelves, and it has a lot of potential to predict biogeochemical – biological (production / respiration pelagic rates) conditions under future climate change scenarios. In spite of that, the authors do not explore this issue, or at least include a discussion paragraph about it in the manuscript. I think that the manuscript would greatly improve with some specific paragraph about their future modeling work, analyzing future biogeochemical consequences of climate change.

Reply: We agree with the comment of the reviewer, coupled physical/biogeochemical models are useful tools to analyse and predict the evolution of biogeochemical fluxes under climate change. The present study is a first step for further investigations of the impact of climate change on the ecosystem, biogeochemical and physical fluxes. We will extend the conclusion section to "conclusion and future works" and will add in this section a paragraph about the impact of climate change on biogeochemical implications in the Gulf of Lion and how future modelling works will help to understand those changes. The results of Herrmann et al. (2014), based on the same biogeochemical model, showed that the water warming could induce an increase in primary production at the scale of the whole northwestern Mediterranean sea under the A2 climate change scenario. Besides, Herrmann et al. (2008) showed a reduction of 90% of shelf-dense water cascading on the Gulf of Lion shelf at the end of the 21st century based on the A2 scenario, suggesting a strong decrease of POC input to the deep sea and an increase of POC transport towards the Catalan Shelf. In future works, we plan to investigate the impact of increasing temperature and stratification also predicted in more recent regional scenarios (RCP8.5 (Soto-Navarro et al., 2020) or SSP5-8.5) on the biogeochemical, physical and air-sea carbon fluxes in the shelf.

"This work represents a first step for further investigations. Along with the coupling with a module describing the dynamics of the carbonate system, a coupling with a module of sediment transport would allow a better representation of the tight link between pelagic and benthic processes occurring in shallow regions. Notably the fluxes of organic carbon and

inorganic nutrients contained within the sediment, taking place at the sediment-water interface during storm-induced resuspension. These events are likely to significantly impact primary production as well as the rate of sedimentation and benthic remineralization on the wave-influenced inner shelf. Moreover, the north-western Mediterranean Sea is subject to multiple global and regional stressors. The ongoing sea warming and increasing stratification of the water column (Somot et al., 2006; Darmaraki et al., 2019) could affect the metabolisms rates and ecosystem dynamics. The results of Herrmann et al. (2014), based on the same biogeochemical model applied on annual future periods (end of the 21$^{st}$ century) under the SRES-A2 climate change scenario, showed that the water warming could induce an increase in primary production, respiration, DOM exudation and bacterial growth rates at the scale of the northwestern Mediterranean Sea. In their scenarios, the net metabolism would not vary significantly and the microbial activity would be enhanced. The future projections performed at the scale of the whole Mediterranean Sea by Lazzari et al. (2014), under SRES-A1B scenarios and different river load conditions for a 20 year period, as well as the transient projections by Moullec et al. (2019) under the RCP8.5 scenario also predicted an increase in rates of primary production and respiration at the end of the 21$^{st}$ century, due to the increase in sea temperature. Lazzari et al. (2014) found a change in the distribution of organic carbon with a decrease in the living organisms particulate biomass and an increase in the dissolved organic carbon inventory. This resulted from a reduction of nutrient availability in response to enhanced stratification. However, based on transient simulations over the entire Mediterranean Sea, the future projections of Richon et al. (2019) (under SRES-A2 scenarios) and Pagès et al. (2020) (under a RCP8.5 scenarios) showed an opposite trend, with, in both studies, a decrease in primary production on the shelf of the Gulf of Lion due a depletion in nutrients (phosphate and nitrate, respectively), by the end of the 21$^{st}$ century. The variability of the response of primary production to climate change in the various modelling studies partly lies in the temperature sensitivity in model equations of primary production as pointed out by Taucher and Oschlies (2011). Indeed the rate of primary production in the model used by Herrmann et al. (2014), Lazzari et al. (2014) and Moullec et al. (2019) is dependent on temperature through an exponential function (Eppley et al., 1972; Eq. S58).  Finally, the increasing stratification could also affect the shelf-slope exchange processes impacting the transport of organic carbon from the Gulf of Lion shelf toward the deeper basin. Herrmann et al. (2008) showed a reduction of 90% of shelf-dense water cascading on the Gulf of Lion shelf at the end of the 21$^{st}$ century based on the A2 climate change scenario. This suggests that the transfer of shelf carbon to the deep sea could be drastically reduced and mostly "re-directed" towards the Catalan shelf. In future works, we plan to contribute to the investigation of the impact of climate change on the biogeochemical, physical and air-sea carbon fluxes by developing high-resolution modelling in this shallow area characterized by complex processes at their various interfaces (sea-air, sediment-water and land-sea)."

I think that the authors should contextualize their manuscript. On one hand, the authors should formulate this manuscript in the context of their own future work, as explained above. And on the other hand, the authors should put their work in the context of previous modeling efforts in the Gulf of Lion (GoL) region. I think that these two points will help to contextualize this manuscript, and to identify its novelty. As it stands right now, the manuscript could be only understood as a good modeling exercise. The authors should explain the novelty of the manuscript in the introduction section. On the other hand, the manuscript will probably benefit from integrating the modelled POC budget on the shelf with estimates from the North Western Mediterranean Open Sea obtained by previous model efforts.

Reply: We agree with the Reviewer comment that a synthesis of previous modelling studies in the Gulf of Lion was missing. We will add a paragraph on previous modelling efforts to the section "1.2 Regional settings" explaining the previous model efforts that have been made in the Gulf of Lion and point out the novelty of the present modelling study.

"Although 3D coupled physical/biogeochemical models are useful tools to analyse the POC dynamics in areas characterized by high spatial and temporal variability, studies based on those models in the Gulf of Lion at pluriannual and shelf scale have remained scarce. Pinazo et al. (1996) investigated the influence of upwelling on primary production on the shelf under various typical wind and Northern Current conditions at a month scale, based on a model with a 4 km horizontal resolution. Tusseau et al. (1998) used a model with a resolution of 1/10° (~11 km) to estimate primary production and nitrate inputs on the shelf and in particular the shelf-slope exchanges, over a year. Auger et al. (2011) analysed and estimated POC deposition at the scale of the shelf over a 4-month period using a 1.5-km resolution model. Campbell et al. (2013) studied the influence of an eddy-induced upwelling on the dynamics of nutrients and phytoplankton using a realistic simulation of the year 2001 with a resolution of 3 km. Based on a coupled simulation of 17 months covering the NW Mediterranean Sea with a horizontal resolution of 1.2 km, Alekseenko et al. (2014) examined the spatial and temporal variability of the stoichiometry of the nutrients and phytoplankton in NW Mediterranean Sea. Other high resolution (400 m) modelling studies focused on the eastern part of the shelf, in particular the Bay of Marseille, investigating the influence of Rhone river and Northern Current intrusions on nutrient and phytoplankton dynamics over the period 2007-2011 (Fraysse et al., 2013; 2014; Ross et al., 2016) as well as the variability of the carbonate system in 2007 (Lajaunie-Salla et al., 2021)."

Besides, as suggested by the Reviewer, we will add in the discussion section comparisons of POC fluxes on the shelf with fluxes estimated for the NW Mediterranean open-sea, in particular in the deep-convection area, in previous modelling studies (Raick et al., 2005; Herrmann et al., 2013; Ulses et al.,2016 and Kessouri et al., 2018) to discuss the fluxes on the Gulf of Lion's shelf at the scale of the western Mediterranean Sea.

"The estimates of the annual NPP obtained on the shelf are slightly higher than the ones estimated with the same biogeochemical model in the deep-sea convection ranging between 150 and 175 g C m$^{-2}$ yr$^{-1}$ (Ulses et al., 2016; Kessouri et al., 2018), also showing weak interannual variability"

And

"Finally, considering the continental shelf area (11 000 km$^2$), our estimate of the net cross-shelf export corresponds to a mean value of 21.7 g C m$^{-2}$ yr$^{-1}$. As a matter of comparison, the export of POC below the euphotic zone in the north-western deep convection area characterized by intense winter mixing was estimated in previous modelling studies between 15 and 30 g C m$^{-2}$ yr$^{-1}$ at 100 m by Guyennon et al. (2015), to be 27.4 ± 6.6 g C m$^{-2}$ yr$^{-1}$ at 100 m depth by Ulses et al (2016), and to 25 g C m$^{-2}$ yr$^{-1}$ at 150 m by Kessouri et al. (2018). The modelling results of Ulses et al. (2016) showed that the surface layer (surface to 100 m depth) of the northern-western Mediterranean open-sea was a sink of organic carbon for surrounding shallow areas and a net source of organic carbon for the western and southern open sea of the western Mediterranean Sea. The POC GoL shelf could thus partly transit by the north-western open-sea before being transferred to the rest of the western and southern open seas in surface layer or in deeper layer after downward export."

Below, I make some general comments about different points of the manuscript. I believe that the manuscript will be substantially improved if the authors address those points.

GENERAL COMMENTS

It is clear that hydrodynamics processes, such as coastal upwelling, cascading, and also river input, control the POC budget in the GoL. In fact, the authors explain the seasonal variability of POC export based on these processes. Thus I would suggest that it would be interesting to include some information about the seasonality of these processes in section "1.2 Regional settings". I think that the potential reader will better understand the POC budget by indicating the seasonality of coastal upwelling, cascading, and strength of the Northern Current. On the other hand, taking into account the key role played by the Northern Current in the offshore export of POC, it would be helpful to include this current in Figure 1.

Reply: The Northern Current will be added to Figure 1. Some clarifications about the seasonality of upwelling, cascading, and flood events will be added to the section "1.2 Regional Settings".

"The Gulf of Lion is bordered on the continental slope by the Northern Current associated with the cyclonic general circulation of the western Mediterranean basin (Petrenko et al., 2008) intensified in winter (Alberola et al., 1995). The gulf is impacted by strong continental winds, which favor dense water formation and cascading events in winter (Durrieu de Madron et al., 2013), and coastal upwellings in summer (Millot, 1990; Fraysse et al., 2014). More occasionally, marine storms blowing from the east, particularly in fall and winter, induce strong along-isobath currents on the shelf, which induce powerful exports at the south-western exit of the gulf (Mikolajczak et al., 2020)."

Further details on the physical processes will be given in the new section 2.1.3 (added to answer the next question) to justify the choice of the section through which the transports are calculated.

L264- 266, Figure 7 and Figure 10. It is not completely clear which sections are considered to calculate the volume transport. How deep are the sections, only to 120m or deeper? The depth of the sections should be indicated in this part.

Reply: We will add a specific paragraph in the Method section "2.1.3 Estimation of water, nutrients and POC transport" to clarify our methodology. The flux is calculated for two layers : (1) a layer from the sea surface to 60 m corresponding roughly to the depth of the nutricline in summer (Kessouri et al., 2017) and (2) a layer from 60 m depth down to the bottom with maximum depth depending on the local bathymetry shown on Figure 1. This will be specified in the new section of the revised manuscript, as follows:

"2.1.3 Estimation of water, nutrients and POC transport

Water, nutrients and POC transport are estimated through sections that close off the Gulf of Lion shelf (see Fig. 1). The water column is each time divided into two parts, above and below 60 m corresponding roughly to the depth of the nutricline in summer (Kessouri et al., 2017). The sections are considered down to the bottom with maximum depth depending on the local bathymetry (Figure 1). The "western" section corresponds to the area known to be responsible for deep export by cascading (sometimes down to the bottom of the basin ~2500 m) during cold winters (Ulses et al., 2008c; Durrieu de Madron et al., 2013). This export is restricted to 300-400 m during mild winters and also during eastern storms, which blow predominantly in fall and produce a downwelling in the Cap de Creus Canyon (Ulses et al., 2008a; Mikolajczak et al., 2020). The other section hereafter named "eastern" for the sake of simplicity is known in the eastern part as an intrusion zone of the Northern Current (Conan et al., 1998), while in the center of the shelf, exchanges with the Northern Current have also been (more rarely) documented (Estournel et al., 2003). It is also the area where the Rhone plume most often exits the shelf under prevailing NW to N wind conditions (Gangloff et al., 2017; Many et al., 2018)."

L348 – I would indicate NEP instead of Net Ecosystem metabolism (also figure 10c).

Reply: We will replace Net Ecosystem metabolism by NEP as suggested in the manuscript and Figure 10c.

L349 - Please indicate Total Community Respiration.

Reply: "community respiration" will be replaced by "Total Community Respiration".

L 351- 353 Taking into account the important contribution of the rivers to POC delivery, I would suggest to explain the maxima of POC river fluxes of Figure 10d.

Reply: Clarification will be added to the manuscript.

"It is noticeable that while the concentration of POC in the river decreases during floods, the important volume of water delivered during such events considerably increases the input of POC to the shelf."

L343 -348 and Table 2. It is not clear the amount of remineralized organic carbon in the GoL, based on the heterotrophic respiration and the remineralization term presented in Table 2. It is not clear if this remineralization term in Table 2 only corresponds to surface sediment. In

this case, it should be indicated in Table 2 legend. Besides, autotrophic and heterotrophic respiration account for more than total respiration, following Table 2. These terms must be clarified as it is kind of confusing right now.

Reply: The total respiration, i.e. 318.1 t C yr-1, in Table 2, accounts for the sum of the autotrophic respiration, i.e. 88.6 t C yr-1, and the heterotrophic respiration, i.e. 229.5 t C yr-1, in the water column. The remineralisation term in Table 2 corresponds to the benthic remineralisation and is not included in the total respiration. We will clarify these terms in the legend of Table 2 and detail the total respiration, by specifying autotrophic and heterotrophic respirations in Table 2 and in the revised text.

L450 "…import of nutrients on the shelf from offshore waters of 450 about 22 10$_4$tN yr-1 for nitrate" following Table 1, it was 22.8 10$_4$ tN yr-1

Reply: This error will be corrected.

L452-453 "the difference between nitrate and phosphate being explained by the very high N: P ratio in Rhone river inputs (approx. 80)". A reference of this high N:P ratio is needed here.

Reply: The value of the N:P ratio in the Rhone river inputs around 80 is based here on the in situ daily data (Mistrals-Sedoo database, http://mistrals.sedoo.fr/MOOSE/) used to force our model at the river mouth (section 2.1.2). References to previous studies (Pujo-Pay et al., 2006; Ludwig et al., 2010; Auger et al., 2011) will be added to the manuscript.

L464 -467 It is not clear the high nutrient import for the winter 2012 -13. The authors should clarify this point. This winter is not a cold winter but winter nutrient concentrations were high, and also there was an intense export of nutrients through the west.

Reply: On the Gulf of Lion's shelf the heat loss is estimated at 201 W m$_2$ for the winter 2012/13, above the average for the mean heat loss over the 5 studied years. Based on the CNRM-RCSM4 model, Somot et al. (2006) identified this winter as one of the 10 winters characterized by heat loss above average over the 33-year period 1980-2013. Moreover they found that winter 2012/13 is one of the 5 winters over the 33-year period showing high dense water formation rates in the NW Mediterranean open-sea. The intense and vertical mixing between surface water poor in nutrients and enriched deep water occurring this winter in the open-sea was responsible for a large supply of nutrients in the euphotic layer. When the deep mixing stopped in March 2013, a southeasterly storm generated a transport of nutrient-enriched offshore waters toward the Gulf of Lion's shelf. The cyclonic circulation induced by the southeasterly storm on the shelf (Ulses et al., 2008) favoured the import of those nutrients in the eastern parts in the whole water column and an export of a part of those nutrients in the western parts, as specified by the Reviewer, mainly in the deep layers. The other part of the nutrients imported during this event in the surface layer is consumed by phytoplankton. We will clarify this point in the revised manuscript.

"In March 2013, the strong nutrient input corresponded to the interaction between offshore deep convection and a south-easterly storm. First, intense vertical mixing produced a nutrient enrichment in the euphotic layer (nitrate concentration up to 8 µmol L$^{-1}$ near the surface vs. 3 µmol L$^{-1}$ over the shelf, see Kessouri et al. (2017)) in the deep convection area from mid-January to early March. Then, when the vertical mixing stopped, a strong easterly storm

advected the surface nutrient-enriched open-sea water onto the shelf. The cyclonic circulation induced by the southeasterly storm on the shelf (Ulses et al., 2008) favoured the import of nutrients in the eastern parts through the surface and deep layers and an export of nutrients in the western parts, mainly in the deep layers. This led to a large net input of nutrients that were partly consumed in the surface layer by phytoplankton."

L508: Principal components could be included in material and methods section

Reply: We understand the comment of the reviewer. However, as we use only the PC analysis shortly in the discussion and only on the NPP variability we think that it is better to describe the method in the section "5.2 Biological production".

L 543 Following Table 2, minimum POC river inputs should be 2014-15 and 2015 -16. I would say that following figure 12, minimum deposition in front of Rhone mouth should be 2014-15 and 2015 -16, and maximum 2012-13 and 2011-12.

Reply: There was an error in the main text. We apologize for this error that will be corrected in section 5.3 of the revised manuscript:

L 545 Higher anomalies of NPP were also simulated for 2012-13 (Figure 12). Could these high NPP anomalies also explain the higher POC deposition in 2012-13?

Reply: We agree with the comment of the Reviewer. The positive anomaly in NPP in 2012/13 could also explain the higher POC deposition estimated for this year. This will be mentioned in the revised manuscript.

L 600 - 601 "Rivers contribute to the POC delivery to the shelf with a mean value of 19 $10^4$ tC yr-1 representing 10% of the NPP, and strong changes induced by floods (72% inter-annual variability)." I would only focus on one main result here, or clarify this sentence. I would suggest to only focus on one thing, I would say the importance of POC from rivers.

Reply: We will only focus on the importance of POC from rivers vs. POC from NPP, as suggested by the Reviewer

L 564 -565 The authors indicate that the intense POC exports during winters 2011-12 and 2012-13 were favoured by the intense cascading and marine storms considering the manuscripts of Durrie de Madron et al (2013) and Bourrin et al. (2015). These references correspond to field observations collected during winter 2012 and March 2011. No other reference about the intensity of cascading and storms events are indicated for the other study years. Would it be possible to include other references with interannual data of cascading and storms events for the entire study years of this manuscript? I mean since 2011 till 2016

Reply: We agree with the comment of the reviewer. We will add a reference to the paper of Mikolajczak et al. (2020) in the manuscript as they discussed the variability of cascading and storm events over the whole period of interest, using the outputs of the meteorological model ECMWF. Based on this study, we will add a small discussion on the interannual variability of the intensity and depth of the off-shelf transport through dense shelf water cascading and storm induced circulation.

**References**

Alberola, C., Millot, C., and Font, J.: On the seasonal and mesoscale variabilities of the Northern Current during the PRIMO-0 experiment in the western Mediterranean-sea, Oceanologica Acta, 18, 163-192. 1995.

Alekseenko E., Raybaud V., Espinasse B., Carlotti F., Queguiner B., Thouvenin B., Garreau P., Baklouti M.: Seasonal dynamics and stoichiometry of the planktonic community in the NW Mediterranean Sea: a 3D modeling approach, Ocean Dynamics, 64:179–207 DOI 10.1007/s10236-013-0669-2, 2014.

Auger, P. A., Diaz, F., Ulses, C., Estournel, C., Neveux, J., Joux, F., and Naudin, J. J.: Functioning of the planktonic ecosystem on the Gulf of Lions shelf (NW Mediterranean) during spring and its impact on the carbon deposition: a field data and 3-D modelling combined approach, Biogeosciences, 8, 3231-3261, https://doi.org/10.5194/bg-8-3231-2011, 2011.

Herrmann, M., Estournel, C., Somot, S., Déqué, M., Marsaleix, P., and Sevault, F., Impact of interannual variability and climate change on dense water cascading in the Gulf of Lions, Continental Shelf Research, 28, 2092-2112, 2008.

Herrmann, M., Diaz, F., Estournel, C., Marsaleix, P., and Ulses, C.: Impact of atmospheric and oceanic interannual variability on the Northwestern Mediterranean Sea pelagic planktonic ecosystem and associated carbon cycle, Journal of Geophysical Research: Oceans, 118, 5792-5813, https://doi.org/10.1002/jgrc.20405, 2013.

Herrmann, M., Estournel, C., Adloff, F., and Diaz, F., Impact of climate change on the northwestern Mediterranean Sea pelagic planktonic ecosystem and associated carbon cycle, Journal of Geophysical Research: Oceans, 119, 5815-5836, 2014.

Kessouri, F., Ulses, C., Estournel, C., Marsaleix, P., Severin, T., Pujo-Pay, M., and Taillandier, V.: Nitrogen and phosphorus budgets in the Northwestern Mediterranean deep convection region, Journal of Geophysical Research: Oceans, 122, 9429-9454, https://doi.org/10.1002/2016JC012665, 2017.

Kessouri, F., Ulses, C., Estournel, C., Marsaleix, P., D'Ortenzio, F., Severin, T., and Conan, P.: Vertical mixing effects on phytoplankton dynamics and organic carbon export in the western Mediterranean Sea, Journal of Geophysical Research: Oceans, 123, 1647-1669, https://doi.org/10.1002/2016JC012669, 2018.

Lajaunie-Salla, K., Diaz, F., Wimart-Rousseau, C., Wagener, T., Lefèvre, D., Yohia, C., Xueref-Remy, I., Nathan, B., Armengaud, A., and Pinazo, C.: Implementation and assessment of a carbonate system model (Eco3M-CarbOx v1.1) in a highly dynamic Mediterranean coastal site (Bay of Marseille, France), Geosci. Model Dev., 14, 295–321, https://doi.org/10.5194/gmd-14-295-2021, 2021.

Ludwig, W., Bouwman, A. F., Dumont, E., and Lespinas, F.: Water and nutrient fluxes from major Mediterranean and Black Sea rivers: Past and future trends and their implications for the basin-scale budgets, Global Biogeochemical Cycles, 24, https://doi.org/10.1029/2009GB003594, 2010.

Mikolajczak, G., Estournel, C., Ulses, C., Marsaleix, P., Bourrin, F., Martín, J., Pairaud, I., Puig, P., Leredde, Y., Many, G., Seyfried, L., Durrieu de Madron, X.: Impact of storms on residence times and export of coastal waters during a mild autumn/winter period in the Gulf of Lion, Cont. Shelf Res., 207, https://doi.org/10.1016/j.csr.2020.104192, 2020.

Millot, C.: The gulf of Lions' hydrodynamics. Continental shelf research, 10, 9-11, 885-894, 1990.

Pujo-Pay, M., Conan, P., Joux, F., Oriol, L., Naudin, J., & Cauwet, G.: Impact of phytoplankton and bacterial production on nutrient and DOM uptake in the Rhône River plume (NW Mediterranean). Marine Ecology Progress Series, 315(3), 43–54. doi:10.3354/meps315043, 2006.

Raick, C., E. J. M. Delhez, K. Soetaert, and M. Gregoire: Study of the seasonal cycle of the biogeochemical processes in the Ligurian Sea using a 1D interdisciplinary model, J. Mar. Syst., 55, 177–203, 2005.

Somot, S., Houpert, L., Sevault, F., Testor, P., Bosse, A., Taupier- Letage, I., Bouin, M.-N., Waldman, R., Cassou, C., Sanchez-Gomez, E., Durrieu de Madron, X., Adloff, F., Nabat, P., and Herrmann, M.: Characterizing, modelling and understanding the climate variability of the deep water formation in the North-Western Mediterranean Sea, Clim. Dyn., 51, 1179–1210, https://doi.org/10.1007/s00382-016-3295-0, 2016.

Ulses, C., Auger, P.-A., Soetaert, K., Marsaleix, P., Diaz, F., Coppola, L., and Estournel, C.: Budget of organic carbon in the North-Western Mediterranean Open Sea over the period 2004–2008 using 3-D coupled physical-biogeochemical modeling, Journal of Geophysical Research: Oceans, 121, 7026–7055, https://doi.org/10.1002/2016JC011818, 2016.

---

## Author Response (AR2)

**Response to the editor, the reviewer 1 and the reviewer 2**

Your revised manuscript was re-evaluated by the previous external reviewers and myself. Both reviewers are happy with your responses and revisions. I agree with the reviewers that you have addressed their concerns and further improved the quality of the manuscript. I am therefore pleased to inform you that your manuscript is accepted for publication pending some minor technical corrections:

1) Maintain the consistency intense used in the abstract section: A mixture of the past tense and present tense are used to describe the results.
2) Define the "GoL" when it first time appears in the abstract session
3) Consider to add a multiplication sign (X) before the number and "$10^4$ t"

All changes have been added to the final version of the manuscript. We thank you very much for your constructive and relevant comments and suggestions.

Sincerely,

Gaël Many